# Aerosol measurements with shipborne sun-sky-lunar photometer and collocated multiwavelength Raman polarization lidar over the Atlantic Ocean

Zhenping Yin[1,2,3], Albert Ansmann[1], Holger Baars[1], Patric Seifert[1], Ronny Engelmann[1], Martin Radenz[1], Cristofer Jimenez[1], Alina Herzog[1], Kevin Ohneiser[1], Karsten Hanbuch[1], Luc Blarel[4], Philippe Goloub[4], Gaël Dubois[4], Stephane Victori[5], Fabrice Maupin[5]

[1]Leibniz Institute for Tropospheric Research, Permoserstraße 15, 04318 Leipzig, Germany
[2]School of Electronic Information, Wuhan University, Wuhan, China
[3]Key Laboratory of Geospace Environment and Geodesy, Ministry of Education, Wuhan, China
[4]Laboratoire d'Optique Amosphérique, Université des Sciences et Technologies de Lille, 59655 Villeneuve d'Ascq, France
[5]Cimel advanced monitoring, Paris, France

*Correspondence to*: Zhenping Yin (zhenping@tropos.de)

**Abstract.** A shipborne sun-sky-lunar photometer of type CE318-T was tested during two trans-Atlantic cruises aboard the German research vessel *Polarstern* from 54°N to 54°S in May/June and December 2018. The continuous observations of the motion-stabilized shipborne CE318-T enabled the first-time observation of a full diurnal cycle of aerosol optical depth (AOD) and column-mean Ångström coefficient of a mixed dust-smoke episode. The latitudinal distribution of the AOD from the shipborne CE318-T, Raman lidar and MICROTOPS II shows the same trend with highest values in the dust belt from 0 ~ 20°N and overall low values in the Southern Hemisphere. The linear-regression coefficients of determination between MICROTOPS II and the CE318-T were 0.988, 0.987, 0.994 and 0.994 for AODs at 380, 440, 500 and 870 nm and 0.896 for the Ångström exponent at 440-870 nm. The root-mean-squared differences of AOD at 380, 440, 500 and 870 nm were 0.015, 0.013, 0.010 and 0.009, respectively.

## 1 Introduction

Aerosols do influence the Earth radiation budget, e.g., by absorption and scattering of solar radiation, and modulate cloud formation and cloud microphysical properties by serving as cloud condensation nuclei (CCN) or ice nucleating particles (INP). Although great progress has been made in aerosol observation technologies and climate modeling in recent years, the uncertainty of aerosol radiative forcing in global climate models is still very large due to our poor understanding of aerosol global distribution and aerosol-cloud interactions (Stocker, 2014).

Most of the current aerosol observations are land-based. Spaceborne aerosol observations are available but most of them work in low Earth orbit, which cannot be used to resolve regional aerosol conditions as a function of time of day. However, the ocean, which covers more than 70 % of the Earth's surface and represents one of the largest natural aerosol sources, can hardly

be monitored by land-based instruments. In addition, marine aerosols, which are generated from the oceanic white cap and bubble bursting, impose significant contributions to the global direct radiative forcing (Satheesh and Moorthy, 2005). Long-range transport of aerosols from the continent plays an important role over the ocean as well, making the aerosol conditions even more complicated. The corresponding measurements of aerosol optical properties with passive remote sensing

instruments can be performed on spaceborne, airborne or shipborne platforms. Spaceborne measurements can provide a global, long-term picture of the aerosol conditions. However, the data retrievals for spaceborne measurements require assumptions about the terrain (Hsu et al., 2013; Sayer et al., 2018), which go along with non-negligible errors. Airborne measurements have a large coverage (Karol et al., 2013), but the cost for each flight is high and the aircraft is sensitive to the weather conditions, which makes it less available for long-term observations. Although shipborne observations are challenging compared to land-

based measurements due to the mobility of the platform and the potential for severe weather conditions, progress about sun photometer technologies has been made over the recent 20 years (Karol et al., 2013; Barreto et al., 2016; Livingston et al., 2003). While first shipborne observations were performed during the NASA Sensor Inter-comparison and Merger for Biological and Interdisciplinary Oceanic Studies (SIMBIOS) (Fargion et al., 1999), which was dedicated to inter-calibration and validation for ocean color satellites, datasets meanwhile span over a long period of time (Smirnov et al., 2002;

Knobelspiesse et al., 2004). The Maritime Aerosol Network (MAN), as a component of the AErosol RObotic NETwork (AERONET) (Holben et al., 2001), is the largest long-term aerosol observation network over the ocean (Smirnov et al., 2009). It has provided unique dataset about aerosol optical depth (AOD) and precipitable water vapor (PWV) over the ocean even from Arctic to Antarctica. The data was largely used in the research about dust transport, satellite retrieval validation and atmospheric correction (Smirnov et al., 2011).

MICROTOPS II is the standard device of MAN. However, it is not dedicated to automatic maritime network observations. At least one operator is required, to point the photometer to the Sun for a while to ensure stable measurements, which makes it less available for continuous, unattended measurements. Moreover, it cannot provide aerosol microphysical properties, including size distribution, scattering phase function and single scattering albedo because of missing sky radiance measurements (Smirnov et al., 2009). Therefore, a shipborne photometer based on the advanced sun-sky-lunar photometery

technology (CE318-T), was developed at Laboratoire d'Optique Atmosphérique (LOA), Lille, France, to cover this gap. This new device has all the capabilities of a land-based CE318-T (Barreto et al., 2016), including measurements of AOD from 340 to 1640 nm, PWV, nighttime AOD and almucantar scans, which are required for the retrieval of aerosol microphysical properties. Therefore, it can be directly incorporated into AERONET. In addition, this instrument will be moved to the Arctic on-board RV *Polarstern* with joining the unprecedented Arctic research project MOSAiC (https://www.mosaic-

expedition.org/). The dataset regarding the Arctic seasonal aerosol conditions will definitely be helpful to quantify human effects on global climate change. But before that, we need to address how the shipborne CE318-T setup behaves, how much influence of the sea spray could bring and how about the uncertainty of the AOD measurements under oceanic conditions.

In order to answer these questions, this instrument was tested in the framework of the OCEANET project (Macke et al., 2010) during the past two RV *Polarstern* cruises, PS113 and PS116. PS113 started at Punta Arenas, Chile on 7 May 2018 and ended

at Bremerhaven, Germany on 11 June 2018. In the case of PS116, RV *Polarstern* departed from Bremerhaven on 11 November 2018 and arrived at Cape Town on 11 December 2018 (see Fig. 1 for the ship tracks). Equipped with sophisticated ground-based instruments, including a portable and automated Raman and polarization lidar system Polly[XT] (Engelmann et al., 2016; Althausen et al., 2009), microwave radiometer, meteorological station, shadowband radiometer, full-sky imager and

MICROTOPS II, it provided a unique opportunity to evaluate the capabilities of the photometer prototype and  collect useful feedback for its future developments.

This paper is organised as follows: In Sect. 2, we give a description of the shipborne CE318-T and other applied instruments and data in this paper. Then in Sect. 3.1, we evaluated the daytime results from the shipborne CE318-T through comparisons with MICROTOPS II and we presented the diurnal measurements of the shipborne CE318-T to validate the nighttime AOD

with collocated Raman lidar measurements. In Sect. 3.2, we present two detailed case studies to evaluate the performance of the shipborne CE318-T under pure marine conditions and during the presence of lofted Saharan dust layers. Furthermore, we will demonstrate the potential of the combination of the shipborne CE318-T measurements and the lidar observations for a detailed characterization of a dust case. Finally, in Sect. 4, summarizing and concluding remarks are given.

**2 Instrumentation**

The instruments of the OCEANET project are dedicated to the investigation of aerosol, cloud, and radiation interactions over the remote Atlantic Ocean and the characterization of contrasts between northern- and southern-hemispheric aerosol and cloud conditions.The OCEANET project started in fall of 2009 (Kanitz, 2012). Nearly all the instruments were mounted on the roof of the OCEANET container except the indoor Polly[XT] lidar. The container was located on the helicopter deck, which is behind the bridge, for these two cruises (see Fig. 2). The MICROTOPS II measurements were conducted on the bridge (see Fig. 2). It

should be noted that the 'anthropogenic' smoke from the funnel of the ship could contaminate the shipborne CE318-T measurements. However, this was a compromise between avoiding strong head winds, sea spray and smoke. Nevertheless, we only found an AOD shift of 0.002 at 500 nm between shipborne CE318-T and MICROTOPS II and this was well within the calibration uncertainty of these instruments. Therefore, the influence of the smoke was negligible for our comparisons.

**2.1 Shipborne CE318-T**

The shipborne CE318-T was developed to enable AOD measurements on mobile platforms and to expand the AERONET coverage to the vast ocean area (Goloub et al., 2017). In principle, the instrument is similar to the traditional CE318-T (Barreto et al., 2016) and has nearly the same steps for installation. The apparatus consists of the optical head, rotational base, control unit, air pumping component, weather stop component, compass and GPS modules (see Fig. 3.C). The optical head was the same like the other land-based CE318-T. The GPS receiver and compass module (SIMRAD HS60) were fixed on the platform

together with the photometer robot to assure the same motions. In order to track the sun continuously over the ship, the photometer will firstly go to the sun with the last information (date, time, geolocation, heading, pitch and roll) from the GPS

receiver and compass module. This can help the photometer point to the sun if the ship does not turn quickly. If the photometer does not see the sun, which can be determined through the digital number from direct sun measurements, the head will be controlled to search the sky at 45º in the left and right horizontal panels. When it detects the sun, the new position will be used to calculate the turning angle of the ship and then to correct the azimuth position for next measurements. When the sun is in

the tracking field of view (~ 10º), the photometer will switch into tracking mode like a regular photometer. However, unlike a conventional CE318-T is, the tracking mode by using the 4-quadrant detector, will keep working to compensate the motions of the ship during all the SUN triplet measurements. It is the same procedure for MOON triplet as well. The air pumping module generates compressed dry-clean air to the collimator to prohibit the contamination of the optical window by ambient sea spray. Meanwhile, we changed the wet sensor (a resistor) by an optical rain sensor to prevent the influence of the strong

corrosion from the sea spray. Furthermore, we added an anemometer to help stop the system, because the robot itself will vibrate when wind speed increases to values above 45 km/h. During the two measurement cruises, however, we chose a limit of 40 km/h to ensure measurements that are unaffected by wind-driven vibrations.

The photometer arrangement is very robust and robotic to conduct 24/7 measurement without special care. The new rain sensor and anemometer worked well even under stormy and rainy weather conditions during the two *Polarstern* cruises. The collected

data was finally transferred to the LOA server for further analysis.

The prototype of the shipborne CE318-T, which was deployed for our study, has 10 channels with nominal wavelengths of 340, 380, 440, 500, 532, 670, 870, 937, 1020, 1064 nm. It can provide AOD values at nine wavelengths and PWV at both daytime and nighttime. It also has the potential of performing almucantar scanning. Further efforts and investigations, such as complex compass data analysis, will be made to utilise these data for the retrievals of aerosol microphysical properties. The

data processing, which we applied, followed the same procedure described in Barreto et al. (2016). In addition, it is required to save the geolocation data along with the AOD since the platform keeps moving all the time.

## 2.2 MICROTOPS II

AOD and PWV measurements were also performed with a handheld MICROTOPS II (Ichoku et al., 2002; Smirnov et al., 2002) within the framework of MAN, which was proceeded by SIMBIOS (Sensor Intercalibration and Merger for Biological

and Interdisciplinary Oceanic Studies). It was calibrated before and after the cruise by NASA Goddard Space Flight Center. This type of MICROTOPS II has 5 channels at 380, 440, 675, 870 and 936 nm.

There are three data quality levels for the AOD both from shipborne CE318-T and from MICROTOPS II: Level 1.0 with no cloud screening, Level 1.5 with cloud screening and Level 2.0 (Level 1.6 for shipborne CE318-T) for cloud screening and quality assurance (Smirnov et al., 2011). We used Level 2.0 (Level 1.6 for shipborne CE318-T) AOD at 380, 440, 500 and

870 nm for our analysis. However, we need to point out that 500 nm AOD from MICROTOPS II database was interpolated with using the Ångström exponent between 440 and 870 nm wavelength.

## 2.3 Polly[XT]

The Raman polarization lidar (Polly[XT]) was continuously operated during the entire cruise. Polly[XT] has two telescopes with diameters of 50 and 300 mm, respectively. There are 12 detection channels connected with these two telescopes, to cover the detection range from near the surface (~120 m) up to 4 km (near-range) and from 800 m to more than 10 km (far-range), respectively. It has 8 far-range channels for wavelengths of 355 nm (total: elastic signal and cross-polarized: filtered by a polarizer), 387 nm, 407 nm, 532 nm (total and cross-polarized), 607 nm and 1064 nm, 4 near-range channels for wavelengths of 355 nm, 387 nm, 532 nm and 607 nm (Engelmann et al., 2016). The signal can be used to retrieve the vertical profiles of volume depolarization ratio at 355- and 532 nm, extinction coefficient at 355- and 532 nm, and backscatter coefficient at 355-, 532- and 1064 nm, which are related to aerosol bulk properties. Hence, particle depolarization ratios at 355- and 532nm and lidar ratios at 355- and 532 nm can be retrieved, which are sensitive to particle size, shape and chemistry properties (Freudenthaler et al., 2009; Baars et al., 2016). The backscatter coefficient $\beta$ and extinction coefficient $\alpha$ are good indicators for particle concentration (Ansmann and Müller, 2005). The lidar ratio S, which is the ratio of extinction to backscatter coefficient, describes the particle absorption ability (Müller et al., 2007; Groß et al., 2011a). Absorbing particles like soot and black-carbon-containing particles have a higher lidar ratio than, e.g., non-absorbing sulfate aerosol particles. Ångström exponent Å (Ångstrom, 1964) which describes the relationship between optical properties (backscatter, extinction) at two wavelengths can be used as an indicator for particle size (Baars et al., 2016; Ansmann et al., 2002). Normally, large particles like dust particles, have a small Å (< 0.5). On the contrary, small particles like biomass combustion aerosols and most continental aerosols, have a larger Å (> 1.0) (Müller et al., 2007; Baars et al., 2016; Eck et al., 1999). Therefore, aerosol layers with different physical and chemical properties, like marine aerosol, dust and smoke, can be characterized based on these retrieving results.

The near-range telescope can suppress the range of incomplete overlap between the laser pulse and the telescope field-of-view to 120 m, which enabled us to capture the aerosol distribution and evolution inside the marine boundary layer (MBL) (Kanitz et al., 2013; Engelmann et al., 2016). In order to avoid any damage of the photon-counting detectors from strong solar radiation, the lidar system was turned off when the solar elevation angle exceeded 70° and the 407 nm channel was turned off routinely at daytime.

In order to calculate the AOD from the lidar observations, the Raman method (Ansmann et al., 1992) and the Klett-Fernald method (Fernald et al., 1972) were utilized for nighttime and daytime measurements, respectively. The Fernald method needs the assumption of a lidar ratio, which is dependent on aerosol types. In our analysis, lidar ratios of 20 sr (20 sr and 20 sr), 50 sr (50 sr and 50 sr) were used for marine aerosols and dust at 355 nm (532 nm and 1064 nm) (Groß et al., 2011a). The assumption about lidar ratio would lead to a maximum relative error of 20 % for AOD, which is dependent on the deviations of lidar ratio for the aerosol layers (Kafle and Coulter, 2013; Hughes et al., 1985). Raman method can achieve better accuracy, because it doesn't need the critical assumption of lidar ratio (Ansmann et al., 1992). However, it can lead to relatively large statistical errors, due to the very weak Raman signal. Therefore, in order to reduce the statistical error to less than 15 %, we

accumulated the signal over 1 hour and used a vertical smoothing window to increase the signal-noise-ratio (Mattis et al., 2004; Groß et al., 2011b).

## 2.4 Supplementary instruments and data sources

Temperature, pressure and relative humidity (RH) profiles were obtained from radiosonde ascents. The radiosondes were launched on board the RV *Polarstern* at 11:00 UTC on each day. For times deviating more than 3 hours from the radiosonde launch, Global Data Assimilation System 1° resolution (GDAS1) meteorology data (Kanamitsu, 1989) was used in the lidar data analysis. This data is processed every three hours per day with a spatial resolution of 1° (latitude, longitude) by an atmospheric model provided by National Centers for Environmental Prediction (NCEP). In addition, the Hybrid Single-Particle Lagrangian Integrated Trajectory (HYSPLIT) model (Draxler, 2011) was used for backward trajectory analysis.

## 3 Results

### 3.1 Validation of shipborne CE318-T

### 3.1.1 Daytime validation with MICROTOPS II

The AOD measurements were conducted with MICROTOPS II, Polly[XT] and shipborne CE318-T simultaneously at daytime and with Polly[XT] and shipborne CE318-T at nighttime. In order to evaluate the reliability and data quality of the shipborne CE318-T, we showed linear regressions between MICROTOPS II AOD and shipborne CE318-T AOD in Fig. 4. Good linear relationship was found between the shipborne CE318-T and MICROTOPS II with $R^2$ (coefficient of determination) of 0.988, 0.987, 0.994 and 0.994 for AODs at 380, 440, 500 and 870 nm, respectively, and of 0.896 for the Ångström exponent. The Ångström exponent is sensitive to the measurement error at clean conditions with AOD less than 0.05. Therefore the scatter in the respective correlation in Fig. 4e is acceptable.

In order to study how the AOD from MICROTOPS II and CE318-T agreed with each other, we used the Bland-Altman plots (Willmott, 1982; Knobelspiesse et al., 2019; Bland and Altman, 1986) to visualize AOD difference ($\Delta AOD = AOD_{CE318-T} - AOD_{MICROTOPS}$) against the AOD mean ($\overline{AOD} = (AOD_{CE318-T} + AOD_{MICROTOPS})/2$, which can clearly display the bias and systematic effects. For this analysis, we only took the data pairs with the 500 nm AOD between 0.04 and 0.2, according to the WMO criteria for traceability (WMO). We used the metric, which is the percentage of $\overline{AOD}$ that falls out of the boundary of the mean difference $\pm 1.96 \times$ the root-mean-squared AOD difference, to quantify the agreement of two measurements. According to the statistical analysis in Knobelspiesse et al. (2019) and Giavarina (2015), the criteria of 5 % for the metric of dropout rate normally can be used to determine the agreement is good or not, if the AODs from two instruments were independent and the AOD difference followed normal distribution. In order to test whether we can take the same criteria, we use the Anderson-Darling test to evaluate the normality and Chi2 test to evaluate independence. The results showed the AOD measurements between CE318-T and MICROTOPS II were independent but the AOD difference did not follow a normal

distribution, which could state potential systematic errors either from MICROTOPS II or from the CE318-T. Under this case, the criteria of 5 % on the dropout rate can only serve as an indicator for agreement.

From Fig. 5, we found small positive biases of 0.0019, 0.0050, 0.0052 and 0.0027 for AODs at 380, 440, 500 and 870 nm, respectively, for the CE318-T compared with MICROTOPS II and the root-mean-squared AOD differences are 0.0149, 0.0128, 0.0099 and 0.0090, respectively. Based on studies of Morys et al. (2001) and Ichoku et al. (2002), the estimated uncertainties of AOD from MICROTOPS II decrease from about 0.02 at 340 nm to about 0.01 at 870 nm, as was derived from comparisons with AERONET master field instruments. This means, we can only validate other instruments to this level of accuracy by taking the MICROTOPS II as the reference. The dropout rate of the AOD difference were 3.80 %, 3.80 %, 7.59 % and 2.53 % at 380, 440, 500 and 870 nm, respectively. These results show that the AODs at 380, 440 and 870 nm from the shipborne CE318-T were in good agreement with MICROTOPS II. AOD at 500 nm was a little bit worse, as the dropout rate exceeded 5 %. But, as we mentioned in Sect. 2.2, the 500 nm AOD from the MICROTOPS II was interpolated from other wavelengths, which might go along with additional uncertainties from assuming a certain Ångström exponent. Overall, we can conclude the daytime capabilities for the shipborne CE318-T under the real marine conditions are as good as of the MICROTOPS II.

### 3.1.2 Nighttime comparisons with Polly[XT]

The shipborne CE318-T has the capability to conduct nighttime measurements. This feature can help us to investigate the diurnal evolution of marine aerosols and dust layers over the ocean. However, this function is more challenging than the daytime measurement as moon tracking is much more sensitive to errors in the leveling adjustment and coordination and orientation data from the compass. Therefore, we need to analyze the accuracy of the nighttime measurements. In Fig. 6, we present the full diurnal measurements from the shipborne CE318-T, Polly[XT] and MICROTOPS II on 26 November 2018. On this day, RV *Polarstern* had just passed Cape Verde and was heading towards Cape Town. A layer of mixed dust and pollution aerosol was observed throughout the whole day. This finding is corroborated by the measurements of the 532 nm volume linear depolarization ratio (Fig. 6c) and a backward trajectory analysis, which is shown in Fig. 7. The backward trajectories show that the air mass that was observed between 1 and 3 km height on 26 November 2018 originated from the Saharan desert and spent six days over Chad and Niger before crossing RV *Polarstern*. All the backward trajectories including the ones for 500 m and 1000 m arrival height crossed the active biomass burning regions two days before arriving RV *Polarstern*. Therefore, the advected dust layer probably took up a large amount of biomass-burning aerosols over central Africa. In order to evaluate the shipborne CE318-T AOD measurements at nighttime, AOD from Polly[XT] was calculated based on the extinction coefficient retrieved with the Raman method (Ansmann et al., 1992). Above 1.5 km to 6 km, the extinction coefficient was taken from the far-range channels and between 0.3 and 1.5 km, data from the near-range channels were used. Below 0.3 km, the extinction coefficient was considered to be constant with height, as displayed in Fig. 8b. Furthermore, we've checked the signal above 6 km and found no additional aerosol layers. The overall relative error of AOD with using this approach was 11-15 %, according to the error analysis (Ansmann et al., 1992; Mattis et al., 2004; Groß et al., 2011b). The time series of AOD can be found in Fig. 6a. The deviation between nighttime shipborne CE318-T and lidar observations of 532 nm AOD was less than 0.03.

Daytime measurements from the shipborne CE318-T are also in good agreement with MICROTOPS II at 11:00 UTC with a deviation of 0.01 and 0.01 for the 500 nm AOD and the Ångström exponent, respectively.

## 3.2 Case studies

In Fig. 9, the latitudinal distributions of AOD at 500 nm (532 nm) from these three instruments are displayed for the data collected during the two RV *Polarstern* cruises PS113 and PS116. In both Fig. 9a and Fig. 9b, all measurements show the same trend with peak values between 0° and 20° N (Kanitz et al., 2013), which is a major outflow region of Saharan dust and biomass-burning aerosols. For PS113, this belt was mainly filled with dust particles, because the Ångström exponent at 440-870 nm was less than 0.4 and AOD at 500 nm exceeded 0.5, which are typical values for Saharan dust (Toledano et al., 2007; Rittmeister et al., 2017). However, for PS116, the air mass in this belt showed a mixture of dust and smoke because the Ångström exponent at 440-870 nm was larger than 1 (Baars et al., 2012). This finding is corroborated by the lidar measurements and backward trajectories, as well. On the contrary, the southern hemisphere contains less anthropogenic aerosols and dust. In most cases, marine aerosol dominated our observations. Nevertheless, lofted biomass burning aerosol from Brazil was observed at 25°S during PS113. This event was also captured by Polly[XT] which revealed a layer top height of 2 km, which is not shown here.

In order to illustrate the aerosol vertical distribution over the Atlantic Ocean and to investigate the behavior of the shipborne CE318-T under different aerosol conditions, we present in the following two subsection the results from shipborne CE318-T, Polly[XT] and MICROTOPS II observations for pure marine conditions and for cases with Saharan dust outbreaks. Detailed analyses were applied based on the diurnal measurements from the shipborne CE318-T and Polly[XT] lidar and daytime measurements from MICROTOPS II.

### 3.2.1 Marine aerosol conditions

On 23 November 2018, RV *Polarstern* was west of Western Sahara and approaching Cape Verde. A northwesterly airflow and clean marine conditions prevailed. The measurements from shipborne CE318-T and Polly[XT] are shown in Fig. 10. According to the 532 nm attenuated backscatter, typical marine aerosol conditions were observed. The 532 nm volume depolarization ratio was less than 0.05 at heights below 1.8 km, which means that the marine boundary layer was dominated by spherical sea salt particles. The backward trajectories shown in Fig. 11 demonstrate that the air mass was mainly carried over the ocean during the past 4 days. Furthermore, no additional aerosol layers were observed above 2 km height. The mean AOD at 532 nm from 08:30 to 11:00 UTC based on shipborne CE318-T measurements was $0.06 \pm 0.01$ and mean Ångström exponent at 440-870 nm was $0.26 \pm 0.03$, These are typical values for marine aerosols, which are dominated by coarse mode sea salt particles (Smirnov et al, 2006). The mean AOD at 532 nm and mean Ångström exponent at 440-870 nm from MICROTOPS II were $0.05 \pm 0.01$ and $0.20 \pm 0.03$, which are in good agreement with the shipborne CE318-T.

Detailed height-resolved aerosol information is displayed in Fig. 12. According to the RH profile (Fig. 12d), the marine layer reached up to about 2 km height. The mean extinction coefficient was 38.5 $Mm^{-1}$, 27.4 $Mm^{-1}$ and 19.2 $Mm^{-1}$ at 355 nm, 532 nm and 1064 nm, respectively, as derived from the Fernald method (Fernald et al., 1972) and assuming a fixed lidar ratio of 20 sr (Groß et al., 2011a). The particle depolarization ratios below 1.6 km height were less than 0.02 at 355 nm and 532 nm. From 1.7 km to 2.0 km height the particle depolarization ratio increased to peak values of 0.09 (0.08) at at 355 nm (532 nm) and RH decreased to 10 % according to the GDAS1 data. These are good indicators for the presence of dried sea salt particles (Haarig et al., 2017; Bohlmann et al., 2018). When RH drops to below 45 %, the spherical marine aerosol particles start to crystallize and become cubic-like in shape. These cubic dry sea salt particles will introduce a relatively strong depolarized signal and lead to the increase of particle depolarization ratio (Haarig et al., 2017).

### 3.2.2 Saharan dust

When RV *Polarstern* approached Cape Verde Islands, a dust outbreak was observed from 27 May to 31 May 2018. The event started with a mixture of dust and smoke above the MBL. Starting on 30 May 2018, the layer ascended to above 1.5 km height and was dominated by pure Saharan dust particles.

The MICROTOPS II, shipborne CE318-T and lidar measurements from 16:00 to 17:00 UTC on 30 May 2018 are displayed in Fig. 13. According to Fig. 13a, the results from the shipborne CE318-T and MICROTOPS II agree well with mean 500 nm AOD of 0.66 ± 0.03 and 0.62 ± 0.02 and a mean Ångström exponent at 440-870 nm of 0.08 ± 0.02 and 0.07 ± 0.01. Both results indicate the presence of a large amount of large dust particles. In Fig 13c, we can see a layer between 0.6 km to 1 km height causing slightly enhanced volume depolarization ratio and a dust layer located between 1.5 and 5 km height with large volume depolarization ratio. Inside the MBL, the volume depolarization ratio was quite low which indicates that the contamination caused by dust sedimentation was small.

In Fig. 14, we present the averaged vertical profiles from the lidar Polly$^{XT}$. The extinction coefficient was retrieved with the Fernald method, assuming lidar ratios of 60 sr (355 nm), 45 sr (532 nm) and 54 sr (1064 nm) for the dust layer and of 25 sr (355, 532, 1064 nm) for the MBL. The lidar ratios at 355 and 532 nm were selected based on nighttime Raman measurements, and the lidar ratio at 1064 nm was obtained from AERONET measurements (Shin et al., 2018). Reference values of the backscatter coefficient were tuned to achieve the best agreement of AOD between lidar and shipborne CE318-T. Inside the MBL, mean extinction coefficients at 355 nm and 532 nm were found to be 245 $Mm^{-1}$ and 241 $Mm^{-1}$ (Fig. 14a), respectively, which is very large compared to the values for pure marine conditions discussed in Sect. 3.2.1. This might be caused by the loading and hygroscopic growth of anthropogenic aerosols. This assumption is corroborated by the backward trajectories in Fig. 15a, because a branch of the backward trajectories arriving at 500 m can be traced back to the European continent. The lofted dust layer extended from 1.5 to 5 km with mean extinction coefficients of 166 $Mm^{-1}$, 161 $Mm^{-1}$ and 159 $Mm^{-1}$ at 355 nm, 532 nm and 1064 nm, and particle depolarization ratios of 0.21 ± 0.05 and 0.31 ± 0.05 at 355 nm and 532 nm. These values are in good agreement with optical properties for pure Saharan dust reported earlier (Groß et al., 2011a; Groß et al., 2011b; Tesche et al., 2009). The backward trajectories shown in Fig. 15b indicate that the air mass observed at 4 km height

originated from Chad, Libya and Sudan, and travelled 6 days from these regions before reaching RV *Polarstern*. A relatively clean layer can be found between the lofted dust layer and MBL with extinction coefficients and particle depolarization ratios of below 25 Mm$^{-1}$ and 0.04, respectively. Therefore, we are convinced that the sedimentation of dust particles was negligible in this case. Above the MBL, from 0.5 km to 1 km height, an aerosol layer was presented that showed enhanced particle

depolarization ratios at 355 nm (532 nm) of 0.11 (0.15). The backward trajectories for this layer were similar to the trajectories shown in Fig. 15a. Therefore, it probably consisted of relatively dry, aged anthropogenic particles or a mixture of dry aged anthropogenic particles and dry sea salt particles.

## 4 Conclusions

Shipborne CE318-T measurements were conducted during two trans-Atlantic RV *Polarstern* cruises together with collocated

observations from Polly$^{XT}$ lidar and independent MICROTOPS II sun photometer. The shipborne CE318-T has a special design to avoid contamination of sea-spray and achieved the goal of automatic measurements over the ocean during the entire 4-5 weeks periods of the two cruises.

From linear regression and Bland-Altman plots, we found the capabilities of the shipborne CE318-T under the real oceanic conditions were as good as the manually operated MICROTOPS II to capture the daytime AOD variabilities. For nighttime

measurements, deviations between the 532 nm AOD observed with Polly$^{XT}$ and the shipborne CE318-T was found to be less than 10 %.

The almucantar scanning option will also be implemented in near future, which will enable the retrieval of aerosol microphysical properties over the ocean. All of these features will significantly increase our potential to characterize marine aerosol distribution over the remote ocean and the impact of continental dust, smoke, and haze outbreaks on the aerosol

conditions far away from the continents, as well as dust transport and dust sedimentation over the less exploited oceans.

*Data availability.* Radiosonde data has been archived in PANGAEA (Schmithüsen, 2019a, b). In addition, Polly$^{XT}$ data and quicklooks of the lidar measurements can be accessed on the PollyNET website (http://polly.rsd.tropos.de/). MICROTOPS II data can be downloaded from the AERONET MAN database (MAN). The shipborne CE318-T data can be accessed through contact with Philippe Goloub (philippe.goloub@univ-lille.fr).

*Author contributions.* ZP performed the lidar data analysis and prepared the manuscript together with AA. PS contributed greatly to the proofreading. MR, CJ and ZP set up the instruments for PS113 and were responsible for the lidar measurements. AH, KO and KH set up the instruments during PS116 and were responsible for the lidar measurements during PS116. PG, LB,

GD, SV and FM built up the shipborne CE318-T and were responsible for the corresponding data analysis. All authors contributed to scientific discussion and in this way to the manuscript preparation.

*Competing interests.* The authors declare that they have no conflict of interests.

*Acknowledgements.* The authors acknowledge funding from ACTRIS under grant agreement no. 262254, ACTRIS-2 under
grant agreement no. 654109 from the European Union's Horizon 2020 research and innovation programme, Labex CaPPA (The CaPPA project (Chemical and Physical Properties of the Atmosphere) and ESA/IDEAS program. We sincerely thank the Alfred Wegener Institute and the RV *Polarstern* crewmembers for their huge support and effort in PS113 and PS116 (acknowledgement no. AWI_PS113_00, AWI_PS116_00). We also appreciate the effort of the AERONET MAN, HYSPLIT teams to provide the additional research data to solid the analysis in the paper. Additionally, Zhenping Yin appreciates the
support from the Chinese Scholarship Council (CSC) to conduct this research under the CSC no. 201706270117.

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

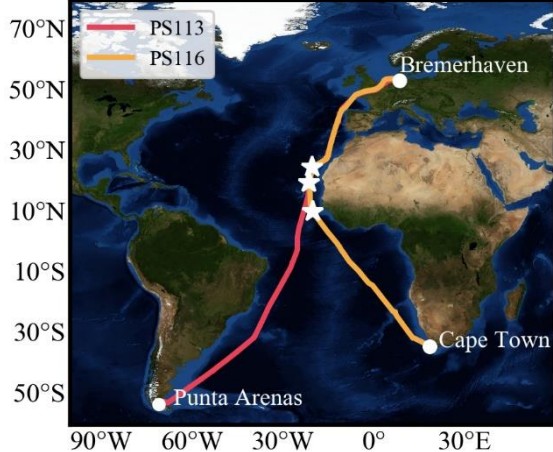

**Figure 1.** Ship tracks for RV *Polarstern* cruises PS113 and PS116. PS113 started from Punta Arenas, Chile on 7 May 2018 and arrived at Bremerhaven, Germany on 11 June 2018. PS116 started from Bremerhaven, Germany on 11 December 2018 and arrived at Cape Town, South Africa on 11 December 2018. White stars mark the location of the case studies presented in Sect. 3.

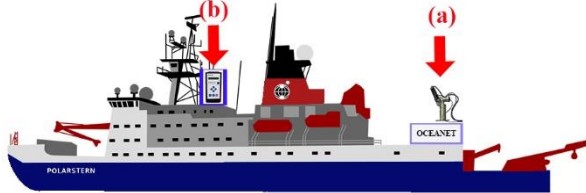

**Figure 2** Photometer and lidar observations aboard RV *Polarstern*. MICROTOPS II observations were performed at site (b). Lidar and shipborne CE318-T observations were conducted at site (a).

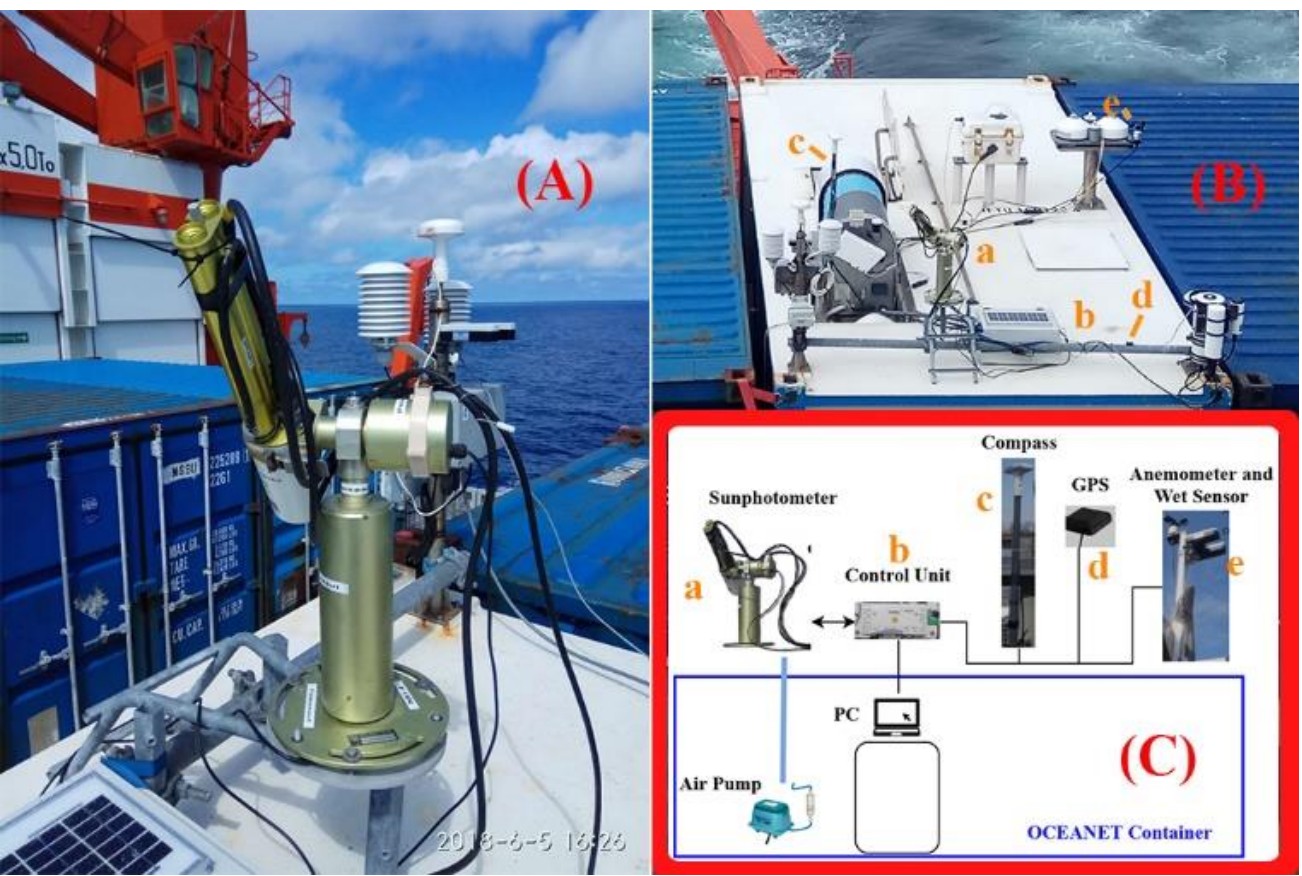

**Figure 3**. Shipborne sun photometer CE318-T taking measurements on the RV *Polarstern* (A), top view of the sun photometer on the top of the container (B) and the sketch of the sun photometer setup (C).

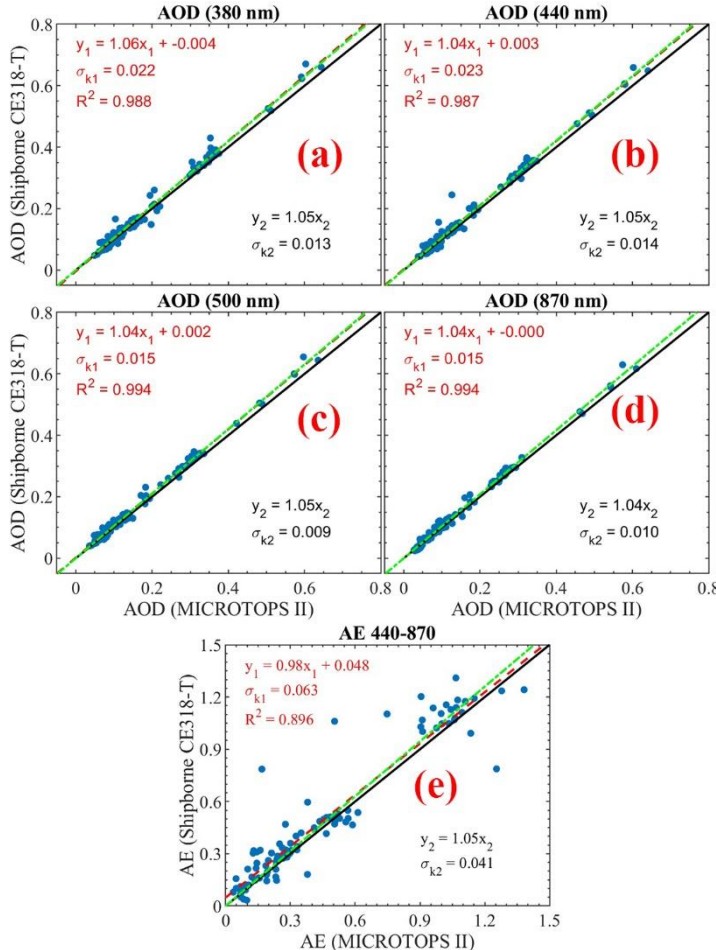

**Figure 4.** Linear regression of AOD (a, b, c, d) and Ångström exponent (e) from the shipborne CE318-T and MICROTOPS II observations. The data points are mean values within a sliding window of 20 min. 115 data pairs are used in this regression. The red dashed line is the regression result with free intercept relationship and the green dot-dashed line represents the regression relationship with forced intercept through 0.

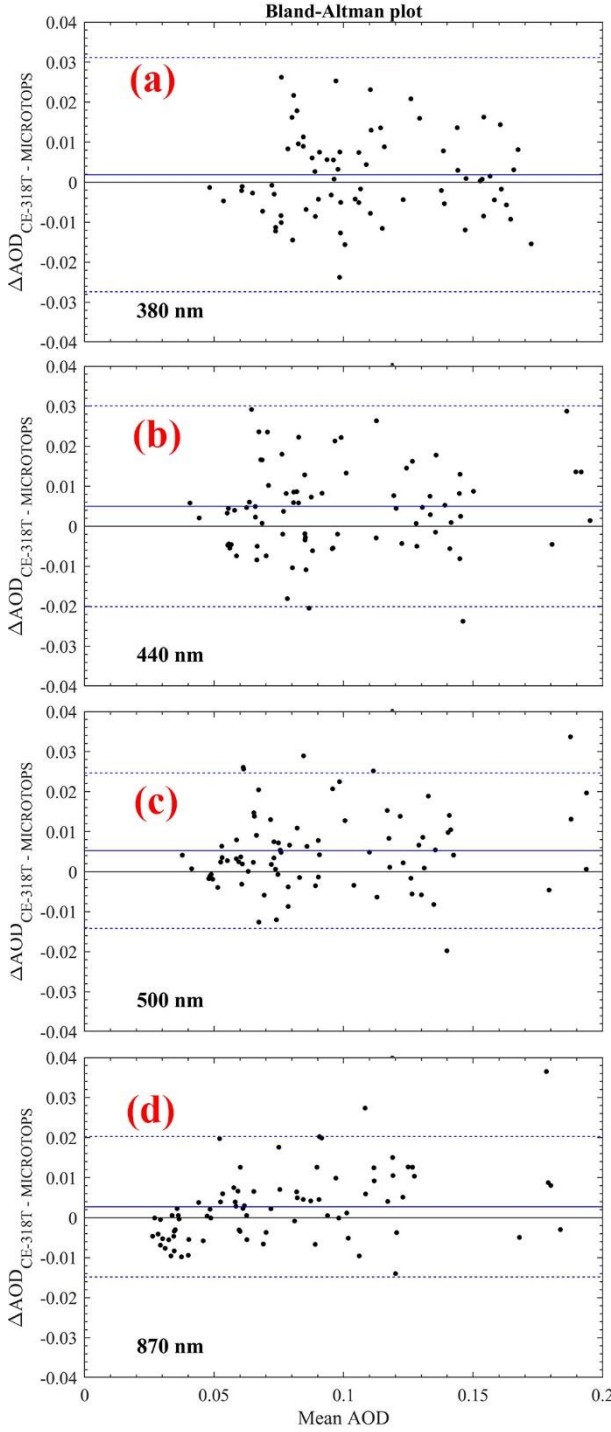

**Figure 5.** Bland-Altman plots for AOD differences with mean AOD (($AOD_{CE-318T}+AOD_{MICROTOPS}$)/2) at 380 (a), 440 (b), 500 (c) and 870 nm (d). The black coloured and blue coloured solid lines represents the zero line and the the mean AOD differences, respectively. Blue coloured dotted lines represent the mean AOD plus/minus the root-mean-squared AOD differences.

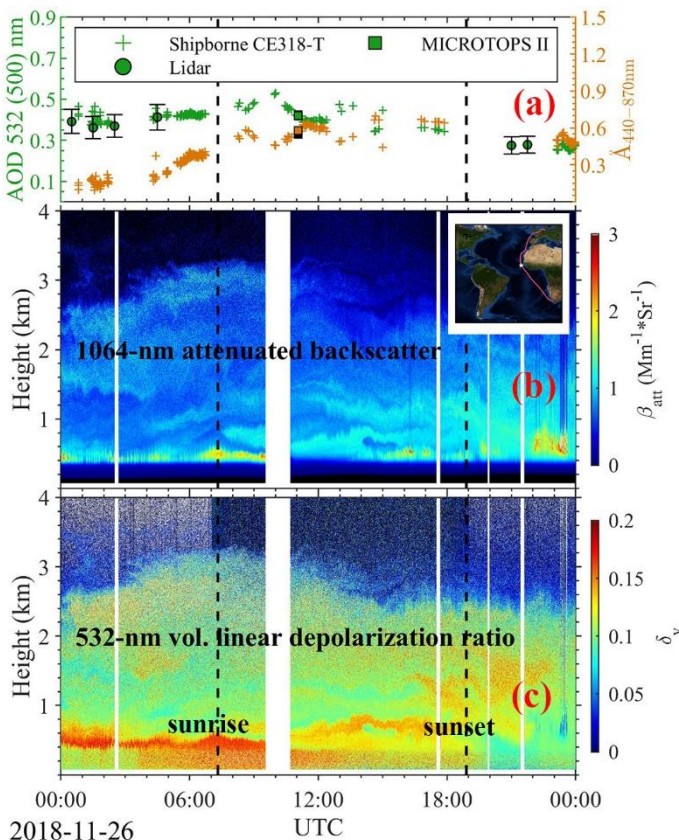

**Figure 6.** Shipborne aerosol observation with CE318-T, MICROTOPS II and Polly[XT] lidar at conditions with a mixture of dust and smoke on 26 November 2018. (a) Comparison of 532 nm AOD from shipborne CE318-T and Polly[XT] lidar observations and 500 nm AOD from MICROTOPS II measurements and Ångström exponent at 440-870 nm obtained from shipborne CE318-T and MICROTOPS II data, (b) mixed layer extended to about 3.5 km height as observed with lidar in terms of 1064 nm attenuated backscatter, and (c) volume depolarization ratio indicating a dust-contaminated MBL. The narrow vertical white stripes are the lidar depolarization calibration periods and the thick white vertical stripe at 10:00 UTC is the routine turn-off time to avoid solar damage at noon.

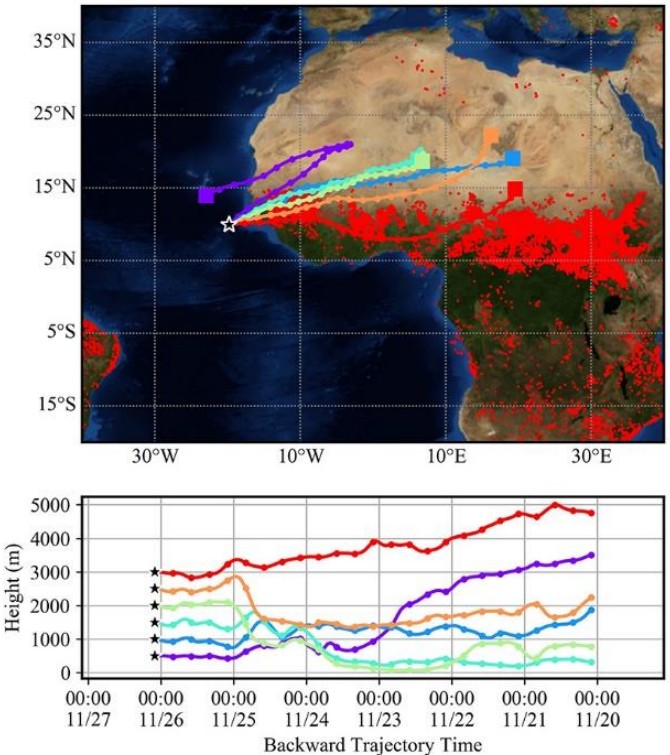

**Figure 7.** NOAA HYSPLIT backward trajectories arriving at RV *Polarstern* (black star with white border, 10.04 °N, 19.82 °W) on 26 November 2018, 02:00 UTC. Red dots are the fire spots detected by MODIS aboard the Terra and Aqua satellites over the period from 20 November to 26 November 2018 (last access: 4 February 2019).

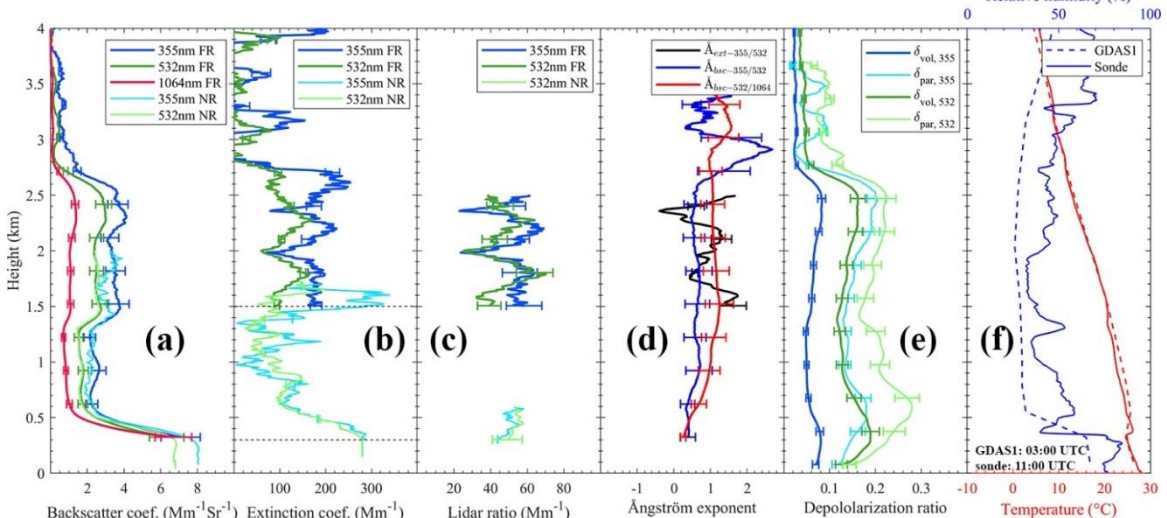

**Figure 8.** Raman lidar observation on 26 November 2018, 02:00-03:00 UTC. (a) Particle backscatter coefficients, (b) particle extinction coefficients (Raman lidar method), (c) lidar ratio, (d) Ångström exponents computed from different wavelengths pair in (a) and (b), (e) volume ($\delta_{vol}$) and particle ($\delta_{par}$) depolarization ratios, and (f) relative humidity (blue) and temperature (red) from radiosonde observations and GDAS1 data.

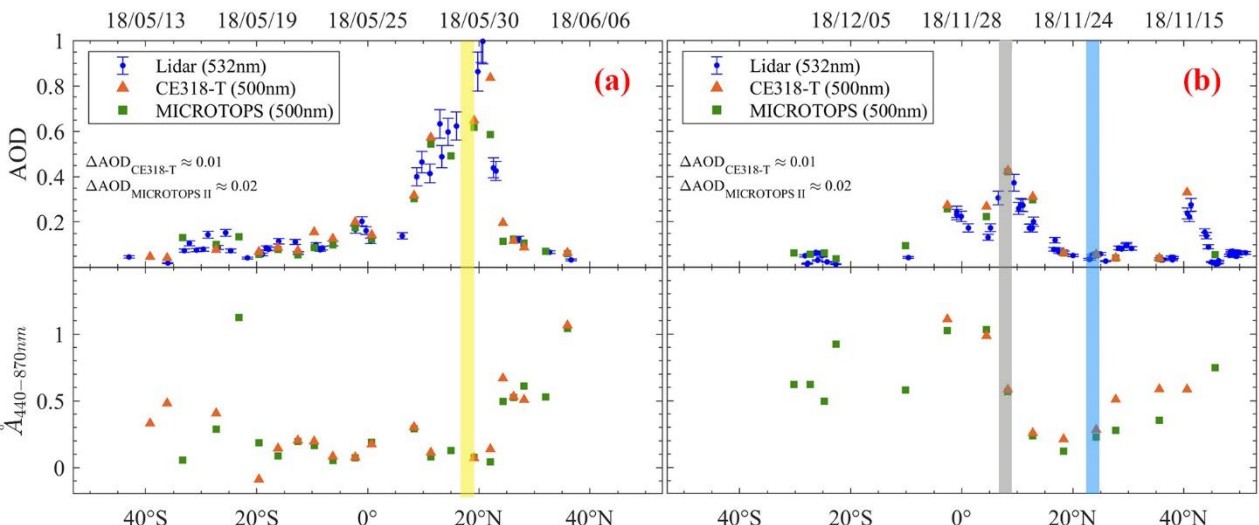

**Figure 9.** (a) Latitudinal distribution of daily mean AOD measured with Polly$^{XT}$ lidar, MICROTOPS II and shipborne CE318-T. Panel (a) and (b) show the results from PS113 and PS116, respectively. The three colored vertical stripes indicate the cases discussed in Sect. 3.1.2 and Sect. 3.2 (yellow: Saharan dust in Fig. 13; grey: diurnal measurements in Fig. 6; blue: pure marine conditions in Fig. 10). Uncertainty in shipborne CE318-T and MICROTOPS II observations were derived according to Smirnov et al. (2009) and Smirnov et al. (2011).

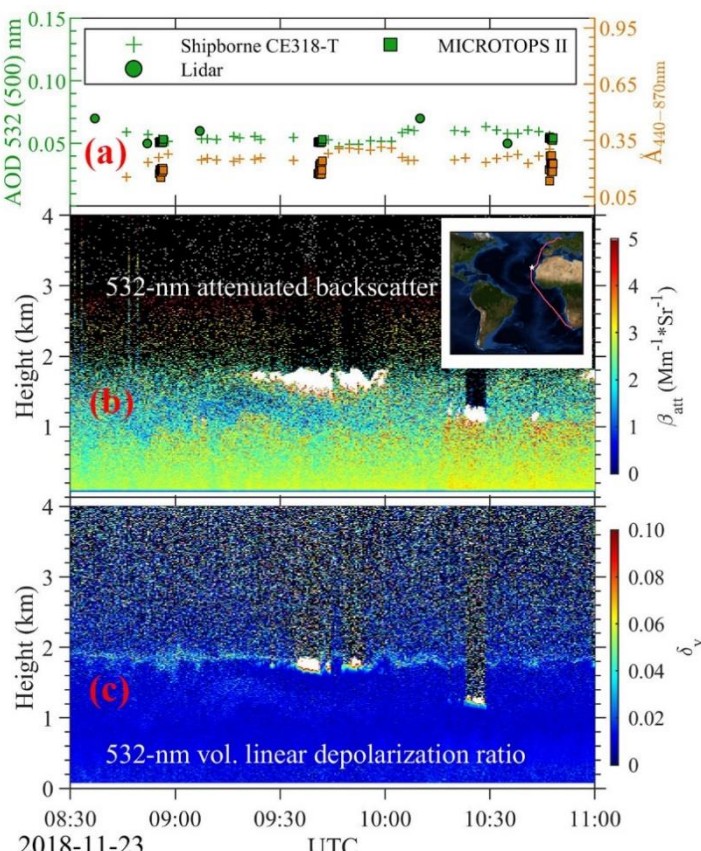

**Figure 10.** Shipborne aerosol observation with the shipborne CE318-T, MICROTOPS II and Polly[XT] lidar at pure marine conditions on 23 November 2018. (a) Comparison of 532 nm AOD measured with shipborne CE318-T and Polly[XT] lidar and 500 nm AOD from MICROTOPS II and Ångström exponent at 440-870 nm from shipborne CE318-T and MICROTOPS II, (b) marine aerosol layer reaching to about 2 km height, partly topped with cumulus clouds (white area), observed with lidar in terms of 532 nm attenuated backscatter, and (c) volume depolarization ratio, indicating pure marine conditions (very low depolarization ratio caused by the spherical droplets as sea salt particle was deliquescent at RH > 70 %) with dried cubic-like sea salt particles at the top (slightly enhanced depolarization ratio) at RH < 45 %.

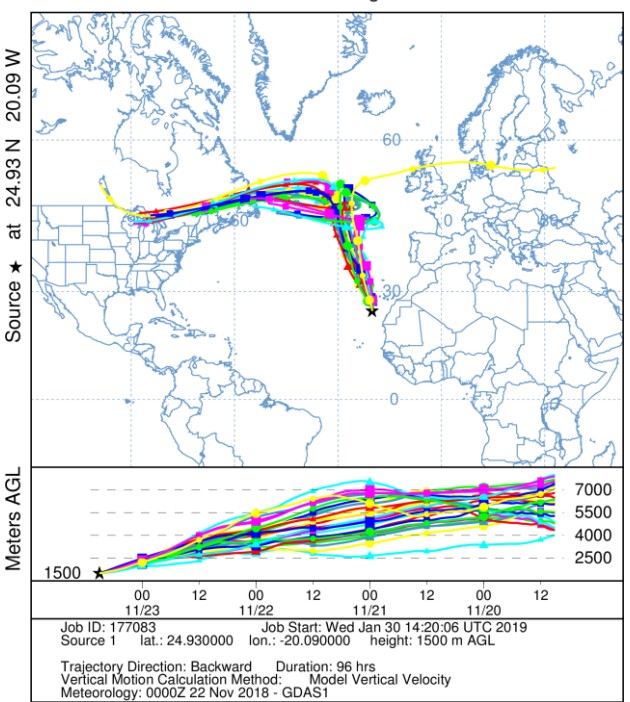

**Figure 11.** Four-day HYSPLIT backward trajectory ensemble arriving at 1500 m height above RV *Polarstern* (black star, 18.41 °S, 32.93 °W) on 23 November 2018, 22:00 UTC.

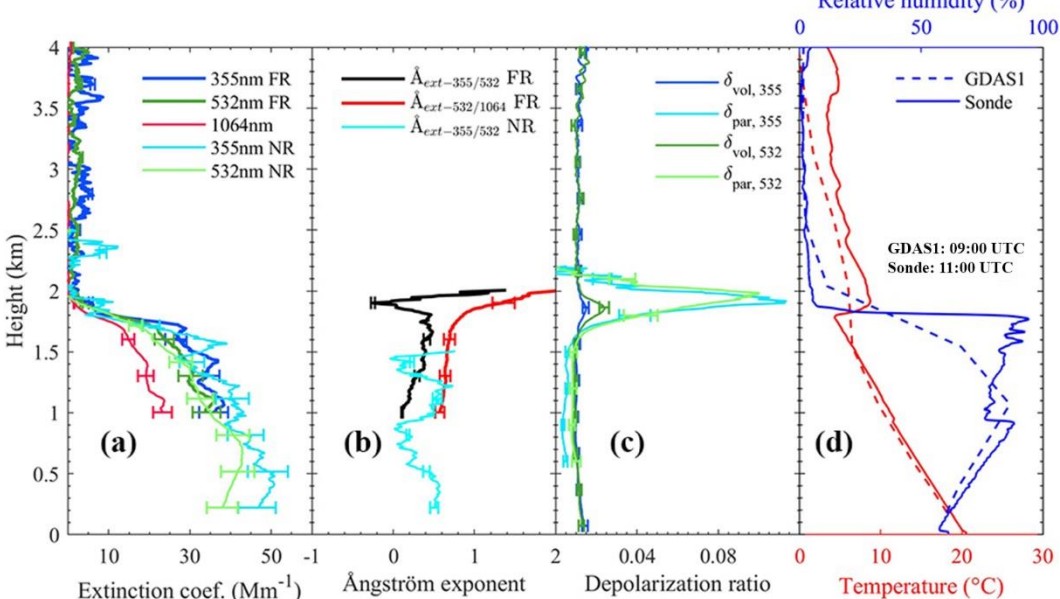

**Figure 12** Height profiles of (a) particle extinction coefficients at 355 nm (blue, FR from far-range signal, NR from near-range signal), 532 nm (green), and 1064 nm (red), (b) Ångström exponents computed from different wavelengths pairs in (a), (c) volume ($\delta_{vol}$) and particle ($\delta_{par}$) depolarization ratios, and (d) relative humidity (blue) and temperature (red). The lidar observations were taken on 23 November 2018, 08:30 – 09:14 UTC. The radiosonde was launched at 11:00 UTC. GDAS1 data for 09:00 UTC are shown for comparison.

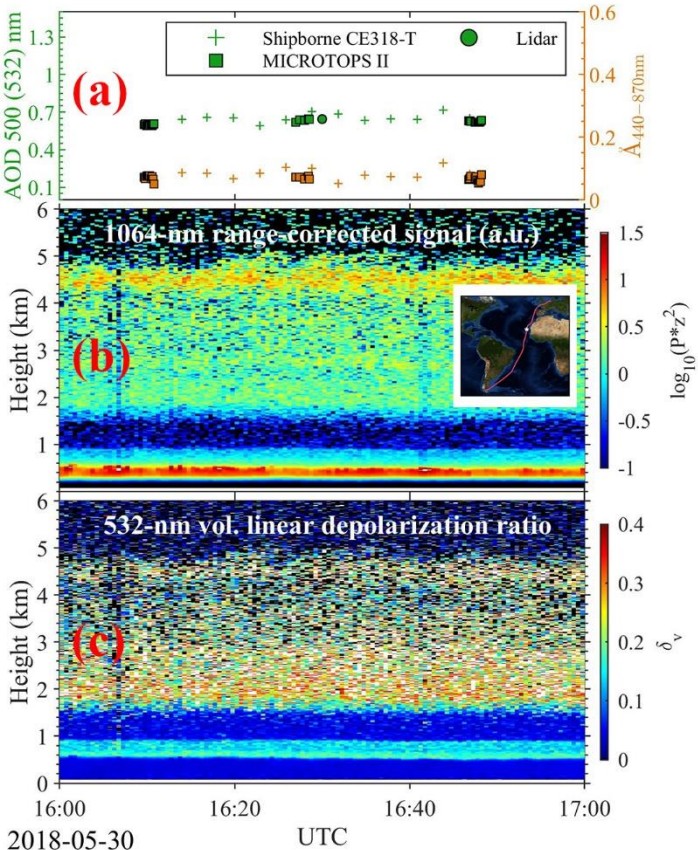

**Figure 13.** Aerosol observation with the shipborne CE318-T, MICROTOPS II and Polly[XT] lidar with strong dust loading on 30 May 2018. (a) Comparison of 500 nm AOD and Ångström exponent at 440-870 nm with shipborne CE318-T and MICROTOPS II, (b) the dust layer extending from 1.5 to 5 km and MBL reaching to 0.6 km, as indicated characterized by the strong range-corrected signal at 1064 nm (red), and (c) volume depolarization ratios indicating the marine layer (low values, blue) and the Saharan dust layer (high values, green and yellow).

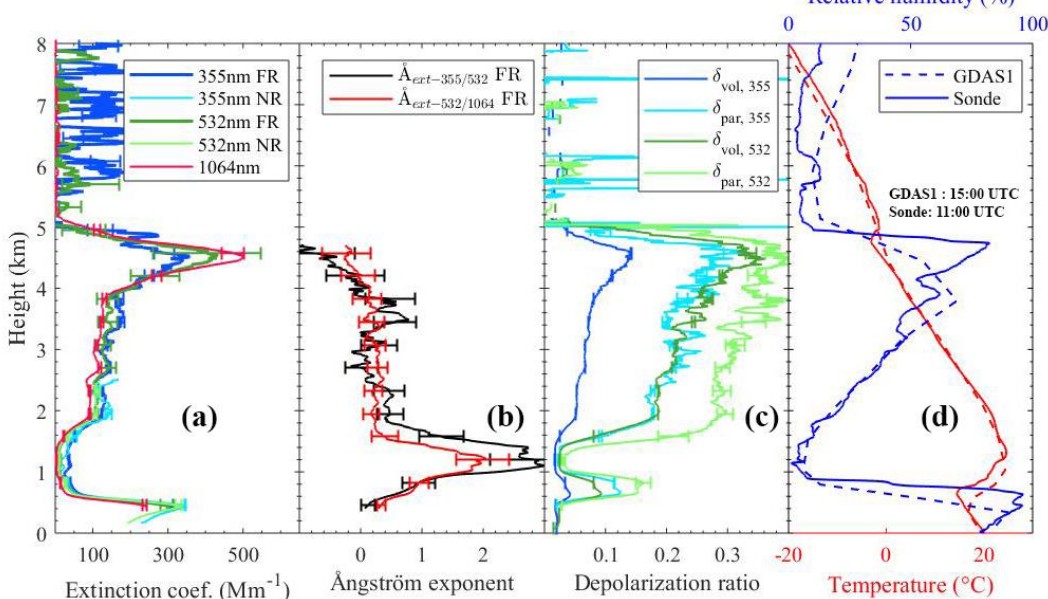

**Figure 14.** Height profiles of (a) particle extinction coefficients at 355 nm (blue, FR from far-range signal, NR from near-range signal), 532 nm (green), and 1064 nm (red), (b) Ångström exponents computed from different wavelengths pairs in (a), (c) volume ($\delta_{vol}$) and particle ($\delta_{par}$) depolarization ratios, and (d) relative humidity (blue) and temperature (red). The lidar observations were taken on 30 May 2018, 16:00 – 16:59 UTC. The radiosonde was launched at 11:00 UTC. GDAS1 data is for 15:00 UTC.

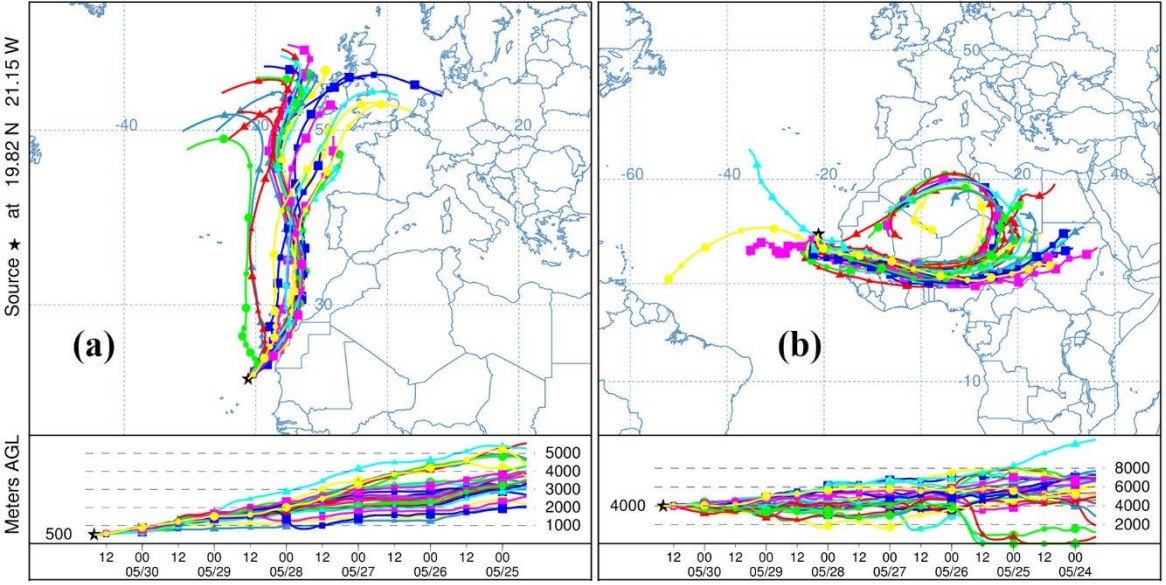

**Figure 15.** Five-day HYSPLIT backward trajectory ensemble arriving at 500 m (a) and six-day HYSPLIT backward trajectory ensemble arriving at 4,000 m (b) height above RV *Polarstern* (black star) on 30 May 2018, 16:00 UTC.