# Peer review of "Aerosol measurements with shipborne sun-sky-lunar photometer and collocated multiwavelength Raman polarization lidar over the Atlantic Ocean"

_Atmospheric Measurement Techniques, 2019_

## Referee Comment (RC1) · Anonymous Referee #1 · 20 Apr 2019

Review of Z Yin et al, "Aerosol measurements with shipborne sun-sky-lunar photometer and collocated multiwavelength Raman polarization lidar over the Atlantic Ocean"

I was excited to have the opportunity to review this manuscript, as I have been anticipating the deployment of an automated ship-borne sun photometer for some time. I commend the authors for their efforts, but am struggling to understand the direction of the paper. Is this intended to be a validation of the sun photometer observations? If so, I find this to be an overly superficial analysis that does not make the case that this instrument is ready for adoption yet. Is this intended to be an examination of the

aerosol state in Atlantic transects? If so, I'm not sure what new information is being presented, nor the implications for our knowledge of the atmosphere.

The conclusion seems to indicate it is the former, so I will proceed as if the purpose of this paper is to demonstrate new observations by the CE318-T. In a general sense, I feel like the following things are not well described and need to be resolved: 1. How different is the shipborne CE318-T from the standard (land based model)? I see that an air pump has been added, along with an anemometer and compass. But how are the compass and GPS data used to help the instrument track the sun? Compensating for ship motion has been an engineering challenge for such instruments for quite some time, so a description of how this has been solved is needed. 2. Does the instrument have a different measurement protocol than the CE318-T? Does it require special data processing? What are the limits of its operation? The anemometer is presumably intended to determine the wind speed and park the instrument when it becomes too rough – at what wind speed does this occur? 3. How accurate do you expect the instrument to be? I would assume calibration and shot noise errors are identical to the land based version, but is there an additional error source due to the difficulties of sun tracking?

These issues would be less relevant if there were some prior literature describing them. As far as I can tell, the only potential description is in the Goloub citation which is a slide presentation for me not appropriate for use as a reference. In any case, that document does not resolve the questions I noted previously. So it falls to this publication to make these things clear.

Regarding your analysis:

Sections 3.1.1 and 3.1.2: ok, these are good. They give an introduction to the comparisons.

Section 3.2.3: an interesting idea to compare this, but I find it very difficult to follow. It is not even clear which data (or times) are associated with day and night – I have to

guess based on the blackout time near solar noon in the lidar data. The conclusions mention issues in this comparison and sensitivity to leveling errors in night time data. But I see no description of these problems in the actual section the conclusion refers to! The conclusion must be supported by previous sections, it is not a place to reveal new information.

Section 3.2: it is unfortunately common in analysis today, but the use of linear regressions and the coefficient of determination to test the hypothesis that two types of measurements agree is statistically unsound. This is especially the case for AOD, which tends to have a log-normal distribution, not a normal distribution that is the basis for linear regression statistics. There is a literature going back decades demonstrating this, including, for example:

Altman, D.G. and Bland, J.M., 1983. Measurement in medicine: the analysis of method comparison studies. Journal of the Royal Statistical Society: Series D (The Statistician), 32(3), pp.307-317.

Bland, J.M. and Altman, D., 1986. Statistical methods for assessing agreement between two methods of clinical measurement. The lancet, 327(8476), pp.307-310. (note this has been cited > 40,000 times and is in the top 100 papers by citation of all time)

Willmott, C.J., 1982. Some comments on the evaluation of model performance. Bulletin of the American Meteorological Society, 63(11), pp.1309-1313.

Or more recently:

Seegers, B.N., Stumpf, R.P., Schaeffer, B.A., Loftin, K.A. and Werdell, P.J., 2018. Performance metrics for the assessment of satellite data products: An ocean color case study. Optics express, 26(6), pp.7404-7422.

And an example of the "Bland-Altman" analysis and the corresponding "Limits of Agreement" can be found here:

Knobelspiesse, K., Tan, Q., Bruegge, C., Cairns, B., Chowdhary, J., van Diedenhoven,

B., Diner, D., Ferrare, R., van Harten, G., Jovanovic, V. and Ottaviani, M., 2019. Intercomparison of airborne multi-angle polarimeter observations from the Polarimeter Definition Experiment. Applied optics, 58(3), pp.650-669.

I gather that the CD318-T and Microtops data do agree well, but I have seen examples of high correlation coefficients masking a bias or other problem with the agreement in data.

Also, why just present AOD(500) and Angstrom? I think a comparison of all available bands would be useful.

In any case, I hope that I have not discouraged the authors from continuing with this important paper. I sincerely hope that this can be published in a form that is useful to the community. I think most of my comments can be addressed with more detailed descriptions, with the exception of section 3.2.

Detailed comments follow:

Page 1 Line 26: "cannot be used to resolve diurnal cycle of the boundary layer" -> perhaps too strong of a statement considering new geostationary observations.

Page 1 Line 30: AERONET-MAN was actually proceeded by a NASA program called SIMBIOS (Sensor Intercalibration and Merger for Biological and Interdisciplinary Oceanic Studies), and in fact inherited the Microtops instruments from SIMBIOS after it ended in 2004. More details about SIMBIOS sun photometry can be found here:

Fargion, G.S., Barnes, R. and McClain, C., 2001. In Situ Aerosol Optical Thickness Collected by the SIMBIOS Program (1997-2000): Protocols, and Data QC and Analysis. NASA/TM-2001-209982, Rept-2001-01672-0, NAS 1.15:209982 https://oceancolor.gsfc.nasa.gov/docs/technical/SIMBIOS-AOT-2000.pdf

Knobelspiesse, K.D., Pietras, C., Fargion, G.S., Wang, M., Frouin, R., Miller, M.A., Subramaniam, A. and Balch, W.M., 2004. Maritime aerosol optical thickness measured by handheld sun photometers. Remote Sensing of Environment, 93(1-2), pp.87-106.

Page 3, line 4: "AOD shift of 0.002" it is not clear which (later) analysis this is referring to

Page 3, lines 8-11: this is where I would have liked more of a description between the differences between the ship-based CD318-T, and the land based version.

Page 3, line 14: What is the wind speed / sea state shut off based on the anemometer? Are there other mechanisms that would shut off the instrument other than clouds and rain?

Page 3, line 19: I think you can't expect that all readers would know what level 1.5 means, this should be explained.

Page 3, line 21: "In the framework of MAN" is vague, are you saying that it is a MAN instrument, or another Microtops that is calibrated, etc. the same way? If the latter, how is this done, specifically?

Page 5, line 10: I'm assuming the +/- values are standard deviations, but it wouldn't hurt to say it.

Page 6, line 19: "free of pollution" Do you mean to say free of fine mode aerosols? I would assume there is a lot of smoke here, but some of it might be natural.

Page 7, line 6: I must assume that between 0:00 and 7:00 UTC is nighttime, but this is never mentioned. What is the local UTC offset, and sun rise and sun set? Seems like an important part of this analysis. It should also be shown in Figure 10, probably easy to indicate with vertical lines at sunrise/set Page 7, lines 13-14: You conclude that the MBL was contaminated by dust and smoke... but this section set out to demonstrate how well day and night sun photometer observations agree, right? Why isn't that discussed instead? And why are the issues mentioned in the conclusions mentioned?

Page 7, lines 18-30: This analysis, and figure 13 are fine, I guess, but considering that the range of locations span two hemispheres and weeks of time, I think this can't be used to demonstrate that Microtopes and Cimel agree. Yes, they both capture the

transition from maritime to African smoke/dust plumes and back again, but those are such large features as to not really illustrate the differences between the instruments.

Page 8, conclusions: see my previous comments, these seem unsupported by the previous material.

Page 11, line 8: This isn't really a citation for MAN. At the very least provide the website address. Or cite the Smirnov 2009 paper instead.

Page 20, line 9: You mention the color red for pure marine conditions in fig 4, on my version of the figure this is marked in blue.
* * *

---

## Referee Comment (RC2) · Anonymous Referee #2 · 30 May 2019

Review of Yin et al, 2019 "Aerosol measurements with shipborne sun-sky-lunar photometer and collocated multiwavelength Raman polarization lidar over the Atlantic Ocean"

General comment

The manuscript by Yin et al, 2019 "Aerosol measurements with shipborne sun-sky-lunar photometer and collocated multiwavelength Raman polarization lidar over the Atlantic Ocean" presents sunphotometery and lidar measurements acquired on board

the ship Polarstern during two transects of the Atlantic Ocean in 2018. The authors show the AOD time series for the two cruises and discuss in more detail three case studies. Unfortunately, I don't think the manuscript is publishable in its current state and needs to be substantially reworked.

My main critique of the manuscript is that the authors do not do enough to show the significance of their contribution. The instruments that the manuscript is based on, namely Cimel318-T, MICROTOPS II and PollyXT lidar, are not new and have been thoroughly discussed in the literature. The data processing techniques are also standard and contain no new developments. The night-time AOD measurements are a bit more novel, but there has been significant development in that area in the past decade with star- and moonphotometers (Perez-Ramirez et al, 2008, 2011, Barreto et al, 2012, Baibakov et al, 2015). Furthermore, the discussion of the three events (background marine aerosols, dust and dust+smoke) while not insignificant, does not lead to any new findings or conclusions. If the emphasis is on the first use of the Cimel318-T on board a ship, then the authors should say so and perhaps include a separate section on how this was accomplished from the engineering point of view. Even so, this by itself might not merit a separate publication in AMT. If the emphasis is on the night-time AOD measurements of marine aerosols, then the authors should spend more time discussing day/night variability, expected variations and the importance of night-time AODs. Finally, if the emphasis is on the actual AOD time series across the Atlantic, including the observed events, the authors should exert more effort to differentiate the dataset from other ship campaigns and coastal area AERONET measurements and otherwise show its significance.

Specific comments

- The literature review of the AOD measurements of marine aerosols and their significance is insufficient – the authors should try to provide a better picture of what has been already done and how their work contributes to the pool of knowledge.

- Since comparing CE318T and PolyXT AOD is a significant part of the analysis, it should be clearly shown how AOD estimated from sunphotometry relates to PolyXT measurements through extinction (I could not find a single formula in the entire paper!) Also, I found practically no discussion of error analysis of the derived quantities.

- There should be a more detailed discussion of the cloud screening techniques and their influence on the results.

- The manuscript should be proofread for multiple English errors scattered throughout the document – I'm only pointing out some of them. Phrases like "slightly enhanced", "quite low" or even simply "large" or "small" need to be quantified. The word "probably" is used too many times and by itself is not adequate when discussing scientific findings.

Technical corrections and minor comments

P1L27: "ARE very challenging"

P2L1: "transport" rather than "transportation"?

P2L10: why only almucantar measurements are discussed? Is your Cimel not capable of Principal Plane measurements?

P2L29: "The instruments . . . are scheduled" is awkward phrasing

P3L6: I feel that calling it a "prototype CE318-T" is misleading since no changes were made to the instrument itself but rather to the platform on which it operates. It would be interesting to see a picture or an engineering diagram of this modification

P3L25: This section should be reworked. The section discusses backscatter and extinction coefficients (presumably derived directly from Raman technique) next to the lidar ratio and (further in the article) Fernald methodology. It should be stated which quantities are measured directly and which are estimated. "Far-range" and "near-range" should be quantified. "Total and cross" should be defined.

P4L7. Not sure what is being meant here by "These intensive parameters are sensi-

tive to particle size, shape and chemistry properties". Which parameters? Angstrom exponent? Particle size and shape are themselves intensive parameters (i.e. are per-particle rather than bulk properties).

P5L4. Why do the authors present the "attenuated backscatter coefficient" rather than the extinction coefficient more relevant to AOD comparisons?

P5L14. The Fernald/Klett method and the choice of the lidar ratio should first be discussed in the methodology section.

P5L18. The last two sentences of the section should use references

P6L4 Which "reference values"? What does it mean "tuned to achieve the best agreement of AOD"? How does it affect your results/comparisons?

References:

Baibakov, K., O'Neill, N. T., Ivanescu, L., Duck, T. J., Perro, C., Herber, A., Schulz, K.-H., and Schrems, O.: Synchronous polar winter starphotometry and lidar measurements at a High Arctic station, Atmos. Meas. Tech., 8, 3789-3809, https://doi.org/10.5194/amt-8-3789-2015, 2015.

Barreto, A., E. Cuevas, B. Damiri, C. Guirado, T. Berkoff, A. J. Berjón, Y. Hernández, F. Almansa, and M. Gil (2012), A new method for nocturnal aerosol measurements with a lunar photometer prototype, Atmospheric Measurement Techniques Discussions, 5 (4), 5527–5569, doi:10.5194/amtd-5-5527-2012

Pérez-Ramírez, D., J. Aceituno, B. Ruiz, F. J. Olmo, and L. Alados-Arboledas (2008), Development and calibration of a star photometer to measure the aerosol optical depth: Smoke observations at a high mountain site, Atmospheric Environment, 42 (11), 2733–2738, doi:DOI:10.1016/j.atmosenv.2007.06.009

Pérez-Ramírez, D., H. Lyamani, F. Olmo, and L. Alados-Arboledas (2011), Improvements in star photometry for aerosol characterizations, Journal of Aerosol Science, 42

(10), 737–745, doi:10.1016/j.jaerosci.2011.06.010.
* * *

---

## Author Comment (AC1) · 14 Aug 2019

**Reply to referees:**

First thing first, we would like to thank the referees for their work and valuable comments. Based on their comments, we believe we have significantly changed and improved the manuscript. The revised version was attached after the replies.

Before we go to the detailed replies, we need to clarify the structure revision. Our idea for this paper are twofold. First, we used the daytime MICROTOPS II data and nighttime lidar data to validate the shipborne CE318-T, which is never done before. In this case, we found results of the shipborne CE318-T are convincing and its setup is robust to enable continuous and unattended measurements over the harsh maritime environment. This is a great benefit for future MOSAiC campaign, because it will on-board RV *Polarstern* again and join the 1 year Arctic measurement campaign, which was dedicated to address our human effects for climate change. On the other hand, we used the shipborne CE318-T data to facilitate the lidar data analysis, for instance, using the AOD measurements from the sunphotometer to constrain the Fernald retrieval for dust cases in Sect 3.2.2, using the Angstroem exponent for Raman retrieval to calculate the vertical aerosol extinction coefficient profile and also using the integrated water vapor to calibrate the lidar water vapor channel to retrieve the vertical water vapor mixing ratio (this part was not included in the paper). These are of great importance to enable operational lidar data analysis for future OCEANET and MOSAiC campaigns as well. We hope with these structure changes, we make the highlights clear.

Besides, we want to correct our descriptions about MICROTOPS II configurations. In the original version, we **made a mistake about the channels, which stated there was a channel at 500 nm**. But after our checking for the manual, **we found there was no 500 nm channel inside**. The AOD at this wavelength from MAN was interpolated with using the Ångström exponent. Therefore, we need to correct this in Sect 2.2. But this correction didn't have influences on the conclusions we've drawn before, because we added the AOD comparisons with other wavelengths at this time and also found good agreement.

**Item by item response to the comments of the reviewers, (comments are in bold)**

**Response to Anonymous Referee #1**

**…, but am struggling to understand the direction of the paper. Is this intended to be a validation of the sun photometer observations? If so, I find this to be an overly superficial analysis that does not make the case that this instrument is ready for adoption yet. Is this intended to be an examination of the aerosol state in Atlantic transects? If so, I'm not sure what new information is being presented, nor the implications for our knowledge of the atmosphere.**

Author response (AR): We feel sorry for the confusing message delivered by this article, and we have revised the structure according to your comments (see the beginning). The motivation for this paper is to validate the shipborne sunphotometer with collocated MICROTOPS II and lidar, and shows its potential for regular lidar data analysis.

Regarding the validation analysis, we revised the whole part with Bland-Altman plot and added more details as being suggested. Detailed replies can be found below and the revisions can be found in the new Sect 3.1 (highlight with color red).

**… 1. How different is the shipborne CE318-T from the standard (land based model)? I see that an air pump has been added, along with an anemometer and compass. But how are the compass and GPS data used to help the instrument track the sun? Compensating for ship motion has been an engineering challenge for such instruments for quite sometime, so a description of how this has been solved is needed. 2. Does the instrument have a different measurement protocol than the CE318-T? Does it require special data processing? What are the limits of its operation? The anemometer is presumably intended to determine the wind speed and park the instrument when it becomes too rough – at what wind speed does this occur? 3. How accurate do you expect the instrument to be?**

AR: The instrument is an extended version of land-based CE318-T. The optical head, GPS and rotation base is still the same. But we added additional air-flow system, compass and anemometer to deal with the harsh maritime environment. The SUN and MOON measurements are still the same but we inserted the GPS data to each line of AOD data. Therefore, each AOD data has a dynamic position tag, compared with the static coordinate for land-based model. That makes the data processing a little bit different, because the additional GPS infomation must be included. At present, this data structure is only incorporated in the PHOTONS database.

For the weather stop system and limits of its operation, we only took into account of rain and wind. For the rain detection, we changed the basic wet sensor (a resistor) for land-based CE318-T with an optical rain sensor. Because the former one didn't work well under the sea conditions due to the corrosion from sea salt. For the limit of wind speed, we set it to be 40 km/h, which is an arbitrary choice. However, this value must be smaller than 45 km/h. Because at that level, as we tested, the robot head will be vibrated by the wind (We can also see this effect in the data).

For the accuracy, we expected it like the AERONET data because the final triplet, including the cloud screening, is the same like AERONET.

Some of the changes regarding this comment was also added in Sect. 2.1 (in red)

**I would assume calibration and shot noise errors are identical to the land based version, but is there an additional error source due to the difficulties of sun tracking?**

AR: Yes, it's possible for the additional error source from the sun tracking, if the ship moves too fast or the wind is too strong. But under that conditions, the upgraded tracking system can compensate large part of the movements and

vibrations. Besides, the triplet check will help filter the data with large variability over 30 s as well. These can screen out measurements under unstable sun tracking due to the vibrations of optical head and tracking base. If there were still possibility for bad data coming from failed sun tracking, we still have additional data from GPS and anemometer to help us exclude the results. And through the comparisons with MICROTOPS, we do see good agreement without any further data quality control.

**These issues would be less relevant if there were some prior literature describing them. As far as I can tell, the only potential description is in the Goloub citation which is a slide presentation for me not appropriate for use as a reference. In any case, that document does not resolve the questions I noted previously. So it falls to this publication to make these things clear.**

AR: More technical details (in red) about the shipborne CE318-T were added in Sect. 2.1, which was listed as below,

"The optical head was the same like the other land-based CE318-T. The GPS and compass modules (SIMRAD HS60) were fixed on the platform together with the photometer robot to assure the same motions. In order to track the sun continuously over the ship, the photometer will firstly go to the sun with the last information (date, time, geolocation, heading, pitch and roll) from the GPS and compass modules. This can help the photometer point to the sun if the ship does not turn quickly. If it does not see the sun, which can be determined through the digital number from direct sun measurements, the head will be controlled to search the sky at $45^0$ in left and right horizontal panel of its first position. When it detects the sun, the new position will be used to calculate the turning angle of the ship and then to correct the azimuth position for the next measurements. When the sun is in the tracking field of view (~ $10^0$), the photometer will switch into tracking mode like a regular photometer. But unlike a conventional CE318-T, the tracking mode by using the 4-quadrant detector, will keep working to compensate the motions of the ship during all the SUN triplet measurements. And for moon triplet measurements, it's also the same procedure. Because of the same triplets. The air pumping module generates compressed dry-clean air to the collimator to prohibit the contamination of optical window by ambient sea spray. Meanwhile, we changed the wet sensor by an optical rain sensor to prevent the influence of the strong corrosion from the sea spray. And we also added an anemometer to help us stop the system, because the robot itself will vibrate as wind speed over 45 km/h. We arbitrarily chose the limit of 40 km/h for our measurements."

**… Section 3.2.3: an interesting idea to compare this, but I find it very difficult to follow. It is not even clear which data (or times) are associated with day and night – I have to guess based on the blackout time near solar noon in the lidar data.**

AR: Thanks for pointing out this. For the new version, we added vertical lines and marks with black color about the sunrise and sunset in Figure 6

**The conclusions mention issues in this comparison and sensitivity to leveling errors in night time data. But I see no description of these problems in the actual section the conclusion refers to! The conclusion must be supported by previous sections, it is not a place to reveal new information.**

AR: We fully agree. It is also too early to draw this conclusion because there are many factors which can contribute to the uncertainty of nighttime measurements, for instance, the weak moon illumination, leveling error and large ship turnings during the night. Based on our measurements, we cannot see which one contributed the most. Therefore, we decided to remove this conclusion.

**Section 3.2: it is unfortunately common in analysis today, but the use of linear regressions and the coefficient of determination to test the hypothesis that two types of measurements agree is statistically unsound. This is especially the case for AOD, which tends to have a log-normal distribution, not a normal distribution that is the basis for linear regression statistics…**

AR: This is a very valuable comment! Triggered by this, we have redone the statistical analysis about the measurement agreements between CE318-T and MICROTOPS II in Sect. 3.1 for the wavelengths of 380, 440, 500 and 870 nm, which was shared by the two instruments. Besides, as suggested, we added the Bland-Altman plot in Figure 5.

Our procedures for the analysis followed 3.C in (Knobelspiesse et al., 2019) and we only use the AOD between 0.04 and 0.2, according to traceability requirements stated in (WMO). This also helps to narrow down the aerosol features to mainly marine aerosols. Our results showed the RMS of the differences between CE318-T and MICROTOPS II is 0.0149, 0.0128, 0.0099 and 0.0090 at 380, 440, 500 and 870. This is in good agreement with the results from land-based comparison by Ichoku et al. (2002), which stated the difference between clean calibrated MICROTOPS II and the AERONET Sun photometer is 0.02 at 380 nm and decreasing down to 0.01 at 870 nm. Detailed comparison results can be found in table 1.

**Table 1 Statistics for Comparison Dataset between CE318-T and MICROTOPS II**

| wavelength (nm) | mean difference (*d*) | RMS of difference (*s*) | upper limit of bound (*d+1.96s*) | lower limit of bound (*d-1.96s*) | independence test | Chi2 test for normal distribution | Percentage of outside of bound |
|---|---|---|---|---|---|---|---|
| 380 | 0.0019 | 0.0149 | 0.0311 | -0.0274 | pass | fail | 3.8 |
| 440 | 0.005 | 0.0128 | 0.0301 | -0.0201 | pass | fail | 3.8 |
| 500 | 0.0052 | 0.0099 | 0.0247 | -0.0142 | pass | fail | 7.59 |
| 870 | 0.0027 | 0.009 | 0.0203 | -0.0148 | pass | fail | 2.53 |

In the comparison, we found the difference of AODs at 4 wavelengths failed the chi2 test for normal distribution, which means there were system errors either in MICROTOPS II or in shipborne CE318-T. Based on the current dataset, we cannot point out which one was responsible for this. However, the RMS difference still showed very good agreement, which is even at the limit of MICROTOPS II measurement error (Ichoku et al., 2002).

**Also, why just present AOD (500) and Angstrom? I think a comparison of all available bands would be useful.**

AR: The idea for showing AOD comparisons only at 500 nm is because this wavelength (532 nm) was commonly used in lidar community. But triggered by your comment, statistical analysis for the AOD at 5 wavelengths were also added in Sect. 3.1.1 and Figure 4 was reproduced for the comparisons at 4 wavelengths.

In general, all of them are in good agreement.

**In any case, I hope that I have not discouraged the authors from continuing with this important paper. I sincerely hope that this can be published in a form that is useful to the community. I think most of my comments can be addressed with more detailed descriptions, with the exception of section 3.2.**

AR: The comments are very informative and friendly. We also hope the paper can be informative and valuable to our community.

**Detailed comments follow:**

**Page 1 Line 26: "cannot be used to resolve diurnal cycle of the boundary layer" ->perhaps too strong of a statement considering new geostationary observations.**

AR: Yes, we agree. And it's not really reasonable to mention the capability of capturing the boundary layer structure as compared with sunphotometer. Because the latter one cannot capture it, either. We revised this sentence with "Spaceborne aerosol observations are available but most of them work in low earth orbit, which can not be used to resolve regional aerosol conditions as a function of time."

**Page 1 Line 30: AERONET-MAN was actually proceeded by a NASA program called SIMBIOS (Sensor Intercalibration and Merger for Biological and Interdisciplinary Oceanic Studies), …**

AR: Thanks for pointing out this and we are very glad to know more details and history about MAN and also SIMBIOS. But since we didn't really focus on SIMBOS, we finally decided to only add a brief mention about this in Sect. 2.2.

'The Maritime Aerosol Network (MAN), which was proceeded by SIMBIO (Sensor Intercalibration and Merger for Biological and Interdisciplinary Oceanic Studies) (Fargion et al., 2001; Knobelspiesse et al., 2004), was working as a component of …'

**Page 3, line 4: "AOD shift of 0.002" it is not clear which (later) analysis this is referring to**

AR: This was concluded by the offset of the linear regression. However, according to Bland-Altman plot, this AOD shift, or the mean difference, at 380, 440, 500 and 870 nm were 0.0019, 0.0050, 0.0052 and 0.0027.

Therefore, we replaced "AOD shift of 0.002" with "mean difference of 0.0019, 0.0050, 0.0052 and 0.0027 at …".

**Page 3, lines 8-11: this is where I would have liked more of a description between the differences between the ship-based CE318-T, and the land based version**.

AR: This whole section was extended according to the previous comments. Please check the revisions in Sect. 2.1.

**Page 3, line 14: What is the wind speed / sea state shut off based on the anemometer? Are there other mechanisms that would shut off the instrument other than clouds and rain?**

AR: We set the maximum wind speed as 40 km/h. This threshold was chosen based on our experience of instrument vibration due to strong winds. This threshold worked quite well for the whole period.

Regarding other shutdown mechanisms, we only took the wind and rain into account, which was enough for us to ensure high quality measurements according to the results from these two Polarstern cruises.

**Page 3, line 19: I think you can't expect that all readers would know what level 1.5 means, this should be explained.**

AR: Thanks for reminding us of this. We added some sentences in Sect 2.1.1 to explain the meaning of different data levels.

"There are three data quality levels for the AOD both from shipborne CE318-T and MICROTOPS II: Level 1.0 with no cloud screening, Level 1.5 with cloud screening and Level 2.0 (Level 1.6 for shipborne CE318-T) for cloud screening and quality assurance (Smirnov et al., 2011). We only use Level 2.0 (Level 1.6 for shipborne CE318-T) for the analysis."

**Page 3, line 21: "In the framework of MAN" is vague, are you saying that it is a MAN instrument, or another Microtops that is calibrated, etc. the same way? If the latter, how is this done, specifically?**

AR: It's an MAN instrument and was sent and calibrated by NASA Goddard Space Flight Center. So it had the same measurement protocol and data quality as other MAN instrument.

In order to make it clear, we change the sentence of "… performed with a handheld MICROTOPS II in the framework of MAN" as "… performed with a handheld MICROTOPS II from NASA Goddard Space Flight Center, which was a standardized MAN instrument.**"**

**Page 5, line 10: I'm assuming the +/- values are standard deviations, but it wouldn't hurt to say it.**

AR: Yes, it's the standard deviations. We kept the conventions applied in many lidar data analysis paper. In order to clarify this, we mentioned in Sect 3.2.1 (with red color) that "0.06 ± 0.01 ('±' means standard deviations)"

**Page 6, line 19: "free of pollution" Do you mean to say free of fine mode aerosols? I would assume there is a lot of smoke here, but some of it might be natural.**

AR: We are sorry about this statement. I just copied and pasted it by mistake in the final editing. We deleted this in the new version.

**Page 7, line 6: I must assume that between 0:00 and 7:00 UTC is nighttime, but this is never mentioned. What is the local UTC offset, and sun rise and sun set? Seems like an important part of this analysis. It should also be shown in Figure 6, probably easy to indicate with vertical lines at sunrise/set**

AR: The labels and white vertical lines for sunrise and sunset were added in Figure 6.

**Page 7, lines 13-14: You conclude that the MBL was contaminated by dust and smoke...but this section set out to demonstrate how well day and night sun photometer observations agree, right? Why isn't that discussed instead? And why are the issues mentioned in the conclusions mentioned?**

AR: Our original idea for the case studies is not only including the comparisons but also showing our analysis about aerosol conditions, since we are most interested in the aerosols. But as we came back to read this part again, we found this could be misleading for the readers to extract the messages. Therefore, we removed the aerosol analysis, and put the conclusion about the comparison in the end to highlight the topic.

Thanks for pointing out this.

**Page 7, lines 18-30: This analysis, and figure 13 are fine, I guess, but considering that the range of locations span two hemispheres and weeks of time, I think this can't be used to demonstrate that Microtops and Cimel agree. Yes, they both capture the transition from maritime to African smoke/dust plumes and back again, but those are such large features as to not really illustrate the differences between the instruments.**

AR: Yes, this figure and the analysis are about the aerosol conditions, not purely about the instrument comparison. Because we thought the latitudinal distribution can give us an overview about the aerosol conditions along the cruise, and tell us that the Atlantic ocean was the playground of dust, smoke and anthropogenic aerosols, not only filled with sea salt.

In the revised manuscript, we moved it to Sect. 3.2.2 to introduce the case studies.

**Page 8, conclusions: see my previous comments, these seem unsupported by the previous material.**

AR: The unsupported conclusions now have been removed. And it was discussed in Sect 3.1.1.

**Page 11, line 8: This isn't really a citation for MAN. At the very least provide the website address. Or cite the Smirnov 2009 paper instead.**

AR: Sorry for the wrong citations. We added the link to the MAN data download page.

**Page 20, line 9: You mention the color red for pure marine conditions in fig 4, on my version of the figure this is marked in blue.**

AR: We have revised the text and used the blue color to represent pure marine conditions in fig 4(now at Figure 6).

**Response to Anonymous Referee #2**

General comments:

**"My main critique of the manuscript is that the authors do not do enough to show the significance of their contribution. …"**

AR: We agree. We firstly explained the reason for this in the beginning of this reply letter. We want to emphasize that our idea for this paper was to validate the shipborne CE318-T and deliver two messages to the readers. First, the current status of this instrument was applicable to capture the aerosol conditions unattended over the vast ocean. And the dataset can be incorporated into lidar data analysis, which is a great advantage for extracting vertical information about marine aerosols.

According to our knowledge, such research was never published before with using the CE318-T on-board a ship and it was also the first time to apply the collocated MICROTOPS II and advanced multiwavelength Raman lidar to validate the daytime and nighttime AOD measurements, respectively.

Besides, we want to add one more point that this instrument will join the unprecedented MOSAiC campaign, which could be important for our better understanding of global climate change. Therefore, our analysis will also lay down the foundation for the our future data analysis.

**Specific comments:**

**"The literature review of the AOD measurements of marine aerosols and their significance is insufficient – the authors should try to provide a better picture of what has been already done and how their work contributes to the pool of knowledge."**

AR: We add two more paragraphs to better explain the status and importance regarding marine aerosols measurements in the introduction (in red).

"Ocean covers more than 70 % of our planet earth, and works as one of the largest natural aerosol sources. Marine aerosols, generated from the oceanic white cap and bubble bursting, impose significant contributions to the global direct radiative forcing (Satheesh and Moorthy, 2005). Meanwhile, the transported aerosols from the continent, which complicates the aerosols conditions over the ocean, also plays an important role. The corresponding measurements by passive remote sensing can be done by spaceborne, airborne or shipborne platforms. Spaceborne measurements can provide a global picture of the aerosol conditions over a long-term. But for the data retrievals in the current stage, they still require assumptions about the terrain (Hsu et al., 2013; Sayer et al., 2018). Airborne measurements have a large coverage (Karol et al., 2013), but the cost for each flight is high and the airplane itself is sensitive to the weather conditions, which makes it less available for long-term observations. Shipborne measurements has been done over a long time (Smirnov et al., 2002; Knobelspiesse et al., 2004). Although it's also challenging compared with land-based

measurements due to the mobility of the platform and severe weather conditions, huge progress of sun photometer technologies (Karol et al., 2013; Barreto et al., 2016; Livingston et al., 2003) has been made over more than 20 years since the beginning of the NASA Sensor Inter-comparison and Merger for Biological and Interdisciplinary Oceanic Studies (SIMBIOS)(Fargion et al., 1999), which is dedicated for intercalibration and validation for ocean color satellites."

Besides, the answer to the question "How their work contributes to the pool of knowledge" can also be found in line 22-30, page 2.

**"Since comparing CE318T and PollyXT AOD is a significant part of the analysis, it should be clearly shown how AOD estimated from sunphotometry relates to PollyXT measurements through extinction (I could not find a single formula in the entire paper!) Also, I found practically no discussion of error analysis of the derived quantities."**

AR: A short answer to this comment. Raman method (Ansmann et al., 1990) has been widely used to obtain aerosol extinction coefficient with Raman Lidar. There are many literatures, documents and presentations (Ansmann et al., 1992; Mattis et al., 2016; Groß et al., 2011b; Whiteman, 1999; Baars et al., 2016) to discuss how it works and how the uncertainty was over more than 20 years. We think doing it again will not bring any new knowledge about this topic and on the contrary, it will make the whole paper cumbersome. Therefore, we decide to present error analysis only in the reply letter. Regarding the revision in the manuscript, we summarized the total relative error for AOD in Sect. 2.3, and also add the error bar to Figure 8

*Explanation about error analysis of aerosol extinction coefficient with Raman method.*

Extinction coefficient can be retrieved from Raman lidar measurements (Eq. 1) (Sect. 4.1(Groß et al., 2011b)).

$$\alpha_{\mathrm{p}}(z, \lambda_0) = (1 + f_p)^{-1} \left\{ \frac{1}{\alpha_m(z, \lambda_0)} \frac{d\alpha_m(z, \lambda_0)}{dz} - \frac{1}{X(z, \lambda_R)} \frac{dX(z, \lambda_R)}{dz} - (1 + f_m)\alpha_m(z, \lambda_0) \right\} (1)$$

In which, $\lambda_0$ and $\lambda_R$ are the wavelengths of emitting laser pulse and Raman scattering, respectively. $f_p$ and $f_m$ are used to describe the wavelength dependence of extinction coefficient from aerosols and air molecules, which are defined as below

$$f_p = \left( \frac{\lambda_0}{\lambda_R} \right)^{\text{Å}} (2)$$

$$f_m = \left( \frac{\lambda_0}{\lambda_R} \right)^{4.085} (3)$$

$\alpha_m$ is the extinction coefficient of air molecules, which can be calculated with high accuracy with the temperature and pressure profile from collocated radiosonde or reanalysis data. $X(z, \lambda_R)$ is the range corrected signal, which is defined as below:

$$X(z, \lambda_R) = CO(z)P(z, \lambda_R)z^2 (4)$$

$C$ is the lidar constant which relates to the system efficiency. $O(z)$ is the overlap function, which states the fraction of the overlap area between laser beam and the field of view of the telescope. The overlap function can be treated as a unity above certain altitude and this altitude is related with the divergence of the laser beam and field of view (FOV) of the telescope. For our lidar system, these altitude are 120 and 800 m for near-range (NR) and far-range (FR) channels, respectively. And $P(z, \lambda_R)$ is the measured signal from Raman channel.

The error of the extinction coefficient can be divided into 2 parts: systematic (**syst**) and statistical (**ran**) error. (Sect. 4.1(Groß et al., 2011b))

$$\Delta\alpha_p^{syst}(z, \lambda_0) = \left|\frac{\partial\alpha_p(z, \lambda_R)}{\partial f_p(z)} \cdot \Delta f_p(z)\right| + \left|\frac{\partial\alpha_p(z, \lambda_R)}{\partial\alpha_m(z, \lambda_R)} \cdot \Delta\alpha_m(z, \lambda_R)\right| (2)$$

$$\Delta\alpha_p^{ran}(z, \lambda_0) = \sqrt{\left(\frac{\partial\alpha_p(z, \lambda_R)}{\partial X(z, \lambda_R)} \cdot \Delta X(z, \lambda)\right)^2 + \left(\frac{\partial\alpha_p(z, \lambda_R)}{\partial C_X(z)} \cdot \Delta C_X^{ran}(z)\right)^2} (3)$$

The systematic error comes from the deviations of the applied Å and molecular extinction coefficient to the truth. For the first part, we took Å from the MICROTOPS II into Raman retrieval, which is 0.6 for the smoke case at November 26 2018. And we finally use the standard deviation (std) of 0.1 to represent the error of Å, which yields a relative error of ~ 0.6 % for extinction coefficient. For the latter part of the systematic error, which came from the variations of molecular density over space and time. According to Mattis et al. (2004), the relative error introduced by the uncertainties with using the temperature and pressure profiles from radiosonde is of the order of 1%-5%. For Raman retrieval, we use the reanalysis data from GDAS1, which has very high accuracy about temperature and pressure compared with radiosonde (Dai et al., 2018). Therefore, the contribution of systematic error should be also at the order of 1%-5%.

The statistical error was contributed by the signal noise (the first term in Eq. 3) and the error of the least-square fit (the second term in Eq. 3, in which, $C_X = \frac{dX(z, \lambda_R)}{dz}$). These two terms can be calculated individually and is related with signal-noise-ratio (SNR). In order to achieve high SNR, we accumulate the nighttime Raman signal within 1 hour and using the smoothing window with 21 bins and 51 bins for NR and FR channels, respectively, to ensure the relative uncertainty was less than 10 % up to 6 km.

We then use the integration of aerosol extinction coefficient to calculate the AOD, with the formula below

$$\tau(\lambda_0, z_0) = \int_0^{z_0} \alpha_p(\lambda_0, z')dz', \ z_0 = 6 \text{ km}$$

where $\tau(\lambda_0, z_0)$ is the aerosol optical depth from ground to $z_0$ at the wavelength $\lambda_0$. Because the extinction coefficient was influenced by the overlap function at near-ground, we then combined the extinction coefficients with the near-range and far-range Raman signal to cover from 120 m to 6 km. And using a constant value at 120 m to represent the aerosol extinction coefficient from ground to 120 m, which caused an error of AOD around 0.005. Besides, we also checked the signal compared with Rayleigh signal to exclude aerosol layers above 6 km for the case on 26 November 2018.

Therefore, in total, the error of AOD from lidar measurements was **11%-15%**, which is 0.04-0.06 at 532 nm, with taking into account of the mean AOD of 0.4.

**"There should be a more detailed discussion of the cloud screening techniques and their influence on the results."**

AR: We need to point out that the data analysis for shipborne CE318-T is nearly the same with AERONET data processing and the only difference is that we need to take care of the geolocation, which was set fixed for AERONET data processing. For the cloud-screening techniques, we follow the triplet stability criterion of AERONET (Smirnov et al., 2000), with $(\tau_{max} - \tau_{min}) < \max\{\tau_{min} * 0.03, 0.02\}$. This is very similar to the cloud-screen techniques applied for MAN data processing (Smirnov et al., 2009), which is $(\tau_{max} - \tau_{min}) < \max\{\tau_{min} * 0.05, 0.02\}$. Although there was manual inspection to further screen out the effects from cloud contamination for MAN data processing, but within our comparisons (see Figure 4), we didn't see any outliers, which states the minor difference of the criteria didn't contribute to strong influences on the comparisons.

**"The manuscript should be proofread for multiple English errors scattered throughout the document – I'm only pointing out some of them. Phrases like "slightly enhanced", "quite low" or even simply "large" or "small" need to be quantified. The word "probably" is used too many times and by itself is not adequate when discussing scientific findings."**

AR: Thanks for pointing out this. The awkward English phrases and expressions have been fixed and we went over the whole manuscript and revise other places as well (in red).

**"P1L27: "ARE very challenging""**

AR: done

**"P2L1: "transport" rather than "transportation"?"**

AR: done and fixed other parts with the same wrong words at the same time.

**"P2L10: why only almucantar measurements are discussed? Is your Cimel not capable of Principal Plane measurements?"**

AR: Regarding the Principal Plane measurements, in theory, it can be done. But it's not our objective for this instrument at the moment, that's why we didn't mention that.

**"P2L29: "The instruments are scheduled" is awkward phrasing"**

AR: It has been revised to "The instruments are dedicated for investigating aerosol cloud and …"

**"P3L6: I feel that calling it a "prototype CE318-T" is misleading since no changes were made to the instrument itself but rather to the platform on which it operates. It would be interesting to see a picture or an engineering diagram of this modification"**

AR: We agree with your argument for the misleading calling, because main new features of this shipborne CE318-T were about the ancillary devices and on the software level. In general, the upgrades were done in three aspects: new compass for orientation measurements, air pumping system for isolating the sea spray and new tracking software to deal with the variable geolocation and orientation. We don't have the engineering diagram of the air pumping system at this moment, because this is an experimental instrument. The pictures about the base and the diagram of the air pumping system was attached below to throw some light on this topic.

[Figure]

**Figure 1 picture of the optical head and tracking base over the OCEANET container and schematic.**

[Figure]

**Figure 2 Schematic of the air pumping system**

**"P3L25: This section should be reworked. The section discusses backscatter and extinction coefficients (presumably derived directly from Raman technique) next to the lidar ratio and (further in the article) Fernald methodology. It should be stated which quantities are measured directly and which are estimated. "Far-range" and "near-range" should be quantified. "Total and cross" should be defined."**

AR: Yes, we agree.

What the lidar can directly measures is the backscatter signal from air molecules and particles.

The quantities of extinction and backscatter coefficients, lidar ratio, particle depolarization ratio and etc., are all retrieved either with Raman method or Fernald method. Raman method only needs the assumption about wavelength dependence of extinction coefficient, for getting all those quantities. This assumption contributes to uncertainties of 10.5% for extinction coefficient at the extreme conditions (Veselovskii et al., 2015). For Fernald method, it needs the

assumption of lidar ratio, which can be erroneous, but it can be constrained with using the collocated sunphotometer measurements (Heese et al., 2010).

Now the Sect. 2.3 was re-worked. The changes triggered by your comments were marked with red color.

**"P4L7. Not sure what is being meant here by "These intensive parameters are sensitive to particle size, shape and chemistry properties". Which parameters? Angstrom exponent? Particle size and shape are themselves intensive parameters (i.e. are per particle rather than bulk properties)."**

AR: Sorry for the misleading expression. The "intensive parameters", namely Ångström exponent and particle depolarization ratio (Amiridis et al., 2015), should be replaced with "intensive properties", which is related to the particle size and shape.

**"P5L4. Why do the authors present the "attenuated backscatter coefficient" rather than
the extinction coefficient more relevant to AOD comparisons?"**

AR: This is because we want to show the temporal-spatial variations of the marine aerosols. Extinction coefficient with Raman method cannot be retrieved in such high resolution with our instrument, due to the weak Raman signal at daytime. The attenuated backscatter itself is also a good indicator if only marine aerosol is present, since the lidar ratio is well characterized for pure marine aerosols, at 25 sr from other measurements (Bohlmann et al., 2018; Haarig et al., 2017).

**"P5L14. The Fernald/Klett method and the choice of the lidar ratio should first be discussed in the methodology section."**

AR: We agree with you. However, Fernald/Klett method has been applied for more than 40 years, there are plenty of articles going into this topic. It's not our objective to show the details about how errors with Fernald/Klett method was calculated, although we have done that for our data analysis and also included the error bars in the averaged profiles in Figure 14. We thought providing some numbers about the error should be sufficient for the general readers who are not lidar expert. Therefore, we added some sentences in Sect 2.3 and referred to the work of (Hughes et al., 1985; Rocadenbosch and Comeron, 1999) to provide the door to know more. See the new Sect. 2.3.

**"P5L18. The last two sentences of the section should use references"**

AR: It has been done.

**"P6L4 Which "reference values"? What does it mean "tuned to achieve the best agreement of AOD"? How does it affect your results/comparisons?"**

AR: The "reference values" means the aerosol backscatter coefficient at the reference height. Normally, if the signal is strong enough to arrive top of troposphere or even stratosphere, we will set the values to 0, because it's nearly aerosol-free at this high altitude. But at daytime and under strong attenuation by dense aerosol plumes, it's hardly to find the aerosol-free height with good SNR.

If the reference values were larger than the true values, the whole profile would be larger and vice versa(Rocadenbosch and Comeron, 1999; Matsumoto and Takeuchi, 1994). Therefore, one has to use other measurements to constrain the reference values. One way to do this is to use the AOD from collocated sunphotometer measurements (Heese et al., 2010). The lidar ratio was taken from nighttime measurements with Raman method, which was valid if the aerosol layer was stable and continuous. In this way, we can retrieve the optical properties of the severe dust storm. The influences of this method was dependent on the lidar ratio and the spatial difference of the detection area between lidar and sunphotometer. For the Saharan dust case in Sect. 3.2, the zenith angle of the sunphotometer is 56° and the optical depth as you can see in Figure 13 is stable over the measurement window of 1 hour. Meanwhile, the retrieved optical properties of dust layer agreed with the results from other literatures (Groß et al., 2011a; Tesche et al., 2009; Rittmeister et al., 2017).

[revised manuscript text omitted]

---

## Author Comment (AC2) · 14 Aug 2019

Thanks for your comments.

The reply letter was attached and the revised manuscript was at the end of the letter.

Please also note the supplement to this comment:
https://www.atmos-meas-tech-discuss.net/amt-2019-132/amt-2019-132-AC2-supplement.pdf

---

## Referee Report (RR1)

Review of revision to "**Aerosol measurements with shipborne sun-sky-lunar photometer and collocated multiwavelength Raman polarization lidar over the Atlantic Ocean" by Z. Yin et al amt-2019-132**

I'm very pleased with both the response to my review (and that of the other reviewer) and the current state of the manuscript. I recommend it for publication.

For what it's worth, a few comments:

1. I find Figure 1 in the response letter (intended for the other reviewer) to be useful. Why not include it in the manuscript itself?

2. For the Bland-Altman test:
*"Besides, we used the metric, which is the percentage of AOD that falls out of the boundary of the mean difference ± 1.96 × the root-mean-squared AOD difference, to indicate the agreement of two 10 measurements."*

Either I'm confused at your description or your intent of this test. Do I understand you're checking if less than 5 percent of the data fall outside the mean+- 1.96rms, where mean and rms are calculated from the comparison of the datasets? If so, I think this might be a way to check for normality, but it does not illustrate the amount of agreement between the data. If, instead, you are checking how much data fall outside the 0+-1.96U range, where U is calculated as the squared sum of the *expected* measurement uncertainties of the pair of instruments, then, yes, this is a metric for whether the agreement is good or not. Perhaps I don't understand, but if that is the case other readers might as well.

3. I believe it is SIMBIO**S** (page 4, line 20)
4. I think some of the grammar, etc. still needs to be cleaned up. The author's use of the term "Besides" is not correct for what I presume is intended. Depending on the context, this might be replaced by "Additionally" (page 3, 7, 10) "Furthermore" (page 4) or removed (page 6, both cases).

---

## Author Response (AR2)

We would like to thank the comments from the associated editor. We put our replies below and attached the manuscript with all the revision markups at the end of the documents.

**Item by item response to the comments of the associated editor, (comments are in bold)**

**1. I find Figure 1 in the response letter (intended for the other reviewer) to be useful. Why not include it in the manuscript itself? [Editor note: I leave this up to you, as the Figure could be useful but your current Figures have similar instrument positioning info.]**

AR: Because we were concerned about the image quality of the right-top panel for the Figure 1 in the response letter. But since it could bring more insight about the instrument, we decided to take it for the manuscript. (See the revised manuscript at the end of the reply letter.)

**2. For the Bland-Altman test:**
**"Besides, we used the metric, which is the percentage of AOD that falls out of the boundary of the mean difference ± 1.96 × the root-mean-squared AOD difference, to indicate the agreement of two measurements."**
**Either I'm confused at your description or your intent of this test. Do I understand you're checking if less**
**than 5 percent of the data fall outside the mean+- 1.96rms, where mean and rms are calculated from the comparison of the datasets? If so, I think this might be a way to check for normality, but it does not illustrate the amount of agreement between the data. If, instead, you are checking how much data fall outside the 0+-1.96U range, where U is calculated as the squared sum of the expected measurement uncertainties of the pair of instruments, then, yes, this is a metric for whether the agreement is good or not.**
**Perhaps I don't understand, but if that is the case other readers might as well. [Editor note: please check and clarify if needed.]**

AR: Sorry for the confusing description. Our idea for using this metric is to evaluate whether the agreement is good or not. In order to quantify the degree of agreement, we followed the idea presented in Knobelspiesse et al. (2019), which used the metric of the percentage of data points falling out of the boundary of d±1.96s (d is mean

AOD difference and s is the root-mean-squared AOD difference). The criteria of 5% dropout rate is the criteria of good agreement under the condition that the AOD difference followed normal distribution. If this was not the case, the criteria could be different (Knobelspiesse et al., 2019; Giavarina, 2015). Therefore, in order to show whether we can use this criteria of 5%, we applied Anderson-Darling test on the AOD difference first to see whether it followed normal distribution. However, in our case, it didn't follow normal distribution for all those tested wavelengths. Therefore, criteria of 5% can only be used as an indicator of the agreement.

In order to clarify this, we revised the description in page 6, line 28 to page 7, line 3, and replace it with the context below

"According to the statistical analysis in Knobelspiesse et al. (2019) and Giavarina (2015), the criteria of 5 % for the metric of dropout rate normally can be used to determine the agreement is good or not, if the AODs from two instruments were independent and the AOD difference followed normal distribution. In order to test whether we can take the same criteria, we use the Anderson-Darling test to evaluate the normality and Chi2 test to evaluate independence. The results showed the AOD measurements between CE318-T and MICROTOPS II were independent but the AOD difference did not follow a normal distribution, which could state potential systematic errors either from MICROTOPS II or from the CE318-T. Under this case, the criteria of 5 % on the dropout rate can only serve as an indicator for agreement."

**3. I believe it is SIMBIOS (page 4, line 20)**

AR: Thanks for pointing out this. I've corrected it in the revised manuscript.

**4. I think some of the grammar, etc. still needs to be cleaned up. The author's use of the term "Besides" is not correct for what I presume is intended. Depending on the context, this might be replaced by "Additionally" (page 3, 7, 10) "Furthermore" (page 4) or removed (page 6, both cases). [Editor note: the Production Office can handle some of these issues, but I encourage you to check as well, in case the meaning of sentences is affected during Production changes for grammar.]**

**Please let me know if you have any questions - I do not forsee needing another round of peer-review after attending to the above.**

AR: Sorry, as a non-native English speaker, grammar is always an issue. In order to correct the wrong English grammar, typos and so on, we've detailed go over the manuscript with the help from a senior scientist. And we've marked all the changes with colored fonts.

**References**

[revised manuscript text omitted]

---

## Author Response (AR3)

Thanks for the comments from the associate editor.

We've revised the manuscript accordingly, please find the new version below with mark-ups.

[revised manuscript text omitted]